# A genome-wide MAGIC kit for recombinase-independent mosaic analysis in *Drosophila*

**Yifan Shen[1,2†], Ann T Yeung[1,2†‡], Payton Ditchfield[1,2§], Elizabeth Korn[1,2], Rhiannon Clements[1,2], Xinchen Chen[1,2#], Bei Wang[1,2], Zixian Huang[1,2¶], Michael Sheen[1,2**], Parker A Jarman[1,2††], Chun Han[1,2*]**

[1]Department of Molecular Biology and Genetics, Cornell University, Ithaca, United States; [2]Weill Institute for Cell and Molecular Biology, Cornell University, Ithaca, United States

*For correspondence:
chun.han@cornell.edu

[†]These authors contributed equally to this work

Present address: [‡]Harvard Medical School, Boston, United States; [§]Center for Community Independence, Revere, United States; [#]Department of Neurobiology, School of Biological Sciences, University of California, San Diego, United States; [¶]Department of Developmental Biology, Washington University School of Medicine, Saint Louis, United States; [**]Warren Alpert Medical School of Brown University, Providence, United States; [††]Yale School of Medicine, New Haven, United States

Competing interest: The authors declare that no competing interests exist.

## eLife Assessment

The study showcases a significant and **important** enhancement of the MAGIC transgenesis method, by extending it genome-wide to all chromosomes. The authors provide **compelling** evidence to demonstrate that the MAGIC mosaic clones can be generated for genes from all, including the 4th chromosome. With this toolkit extension, the method is set to complement the classical FRT/Flp recombination system for gene manipulation in flies.

**Abstract** Mosaic analysis has been instrumental in advancing developmental and cell biology. Most current mosaic techniques rely on exogenous site-specific recombination sequences that need to be introduced into the genome, limiting their application. Mosaic analysis by gRNA-induced crossing-over (MAGIC) was recently developed in *Drosophila* to eliminate this requirement by inducing somatic recombination through CRISPR/Cas9-generated DNA double-strand breaks. However, MAGIC has not been widely adopted because gRNA markers, a required component for this technique, are not yet available for most chromosomes. Here, we present a complete, genome-wide gRNA-marker kit that incorporates optimized designs for enhanced clone induction and more effective clone labeling in both positive MAGIC (pMAGIC) and negative MAGIC (nMAGIC). With this kit, we demonstrate clonal analysis in a broad range of *Drosophila* tissues, including cell types that have been difficult to analyze using recombinase-based systems. Notably, MAGIC enables clonal analysis of pericentromeric genes, deficiency chromosomes and in interspecific hybrid animals, opening new avenues for gene function study, rapid gene discovery, and understanding cellular basis of speciation. This MAGIC kit complements existing systems and makes mosaic analysis accessible to address a wider range of biological questions.

## Introduction

Mosaic animals containing genetically distinct populations of cells in the same organism are useful for in vivo studies of complex biological processes. For this reason, techniques that can generate genetically labeled mosaic clones have been utilized in both vertebrates and invertebrates to study tissue-specific functions of pleiotropic genes, developmental timing, cell lineages, cell proliferation, neural wiring, and many other biological phenomena (*Germani et al., 2018*; *Xu and Rubin, 2012*; *Griffin et al., 2014*). The most popular mosaic techniques rely on site-specific recombination systems, such as FRT/Flp (*Golic and Lindquist, 1989*; *Xu and Rubin, 1993*) and LoxP/Cre (*Henner et al., 2013*;

*Zong et al., 2005*), to induce somatic recombination between homologous chromosomes. Such techniques require the introduction of recombination sites to specific locations in the genome and thus cannot be applied to unmodified chromosomes. To overcome this limitation, we recently developed a recombinase-independent mosaic technique called mosaic analysis by gRNA-induced crossing-over (MAGIC; *Allen et al., 2021*). In MAGIC, the CRISPR/Cas9 system generates double-strand breaks (DSBs) at a predefined genome location to induce homologous recombination in precursor cells during S/G2 phase. Subsequent chromosomal segregation during mitosis can result in clones homozygous for the chromosomal segments distal to the crossover site (*Figure 1—figure supplement 1A*). Two variants of this technique in *Drosophila*, positive MAGIC (pMAGIC) and negative MAGIC (nMAGIC), label the resulting homozygous clones by the presence and absence of fluorescent markers, respectively (*Figure 1—figure supplement 1B*; *Allen et al., 2021*). Like FRT/Flp-based techniques, MAGIC enables characterization of homozygous clones of lethal mutations in otherwise heterozygous animals (*Allen et al., 2021*; e.g. *Figure 1—figure supplement 1C*).

However, unlike FRT/Flp-based techniques, MAGIC does not require prior genetic modification of the test chromosome. Thus, it can potentially be used on any chromosome and have much wider applications. Foremost, mutations of diverse natures have been established for most *Drosophila* genes (*Thibault et al., 2004*; *Hacker et al., 2003*; *Bellen et al., 2011*; *Staudt et al., 2005*; *Bellen et al., 2004*; *Yamamoto et al., 2014*), and thousands of deficiency strains harbor deletions that collectively uncover 98.4% of the *Drosophila* genome (*Cook et al., 2012*). However, most of these mutant chromosomes cannot be analyzed by traditional mosaic techniques due to the lack of FRT sites or incompatibility with the FRT/Flp system. Although FRT sites can be introduced onto mutant chromosomes through genetic recombination, this process is labor-intensive and time-consuming and thus is impractical at a large scale. In contrast, MAGIC can theoretically be applied to any existing stock, including those from classical mutagenesis screens and deficiency libraries, allowing convenient genome-wide mosaic screens. In addition, genes located more proximal to centromeres than existing FRT sites cannot be analyzed by FRT/Flp techniques. In comparison, MAGIC can potentially be used to study these genes because the crossover site in MAGIC can be flexibly defined by users. Lastly, given that MAGIC is compatible with wild-derived chromosomes (*Allen et al., 2021*), it may be able to generate homozygous clones of a chromosome derived from a single species in an interspecific hybrid animal, allowing the study of species-related cell-cell interactions.

Despite these potentials, MAGIC has not been widely adopted by the *Drosophila* community. A major barrier is the lack of gRNA-marker transgenes on most chromosomal arms, which are necessary for DSB induction and fluorescent clone labeling (*Allen et al., 2021*). In addition, the existing gRNA markers suffer from several limitations, including low frequency of clone induction, weak labeling of pMAGIC clones, and suboptimal visualization of nMAGIC clones. Because of these reasons, MAGIC has only been successfully applied to a few genes in a limited number of *Drosophila* tissues (*Allen et al., 2021*; *Chen et al., 2025*).

To overcome these limitations, here we first optimized gRNA-marker designs to improve clonal induction, the brightness of pMAGIC clones, and visualization of nMAGIC clones. Then we generated pMAGIC and nMAGIC gRNA markers for all chromosomal arms and characterized their ability to generate clones. Using this kit, we demonstrate mosaic analysis of centromere-proximal genes, deficiency chromosomes, chromosomes derived from different *Drosophila* species in interspecific hybrid animals. This kit allows optimal clone induction in diverse cell and tissue types and should be useful for studying a wide range of biological processes.

## Results
### New gRNA-marker designs improve pMAGIC and nMAGIC

MAGIC relies on a gRNA-marker dual-transgene (*Figure 1A*, *Figure 1—figure supplement 1B*) inserted in a specific chromosomal arm to both induce and visualize clones homozygous for this arm (*Allen et al., 2021*). The gRNA part of the construct expresses two gRNAs ubiquitously to target a pericentromeric location on this arm. Two gRNAs targeting two sequences that are close to each other, instead of a single gRNA, are used to increase the probability of DSBs and thus the clone frequency. The marker part in pMAGIC utilizes a ubiquitously expressed Gal80 to prevent Gal4-dependent labeling of heterozygous and homozygous cells for the gRNA-marker, while allowing

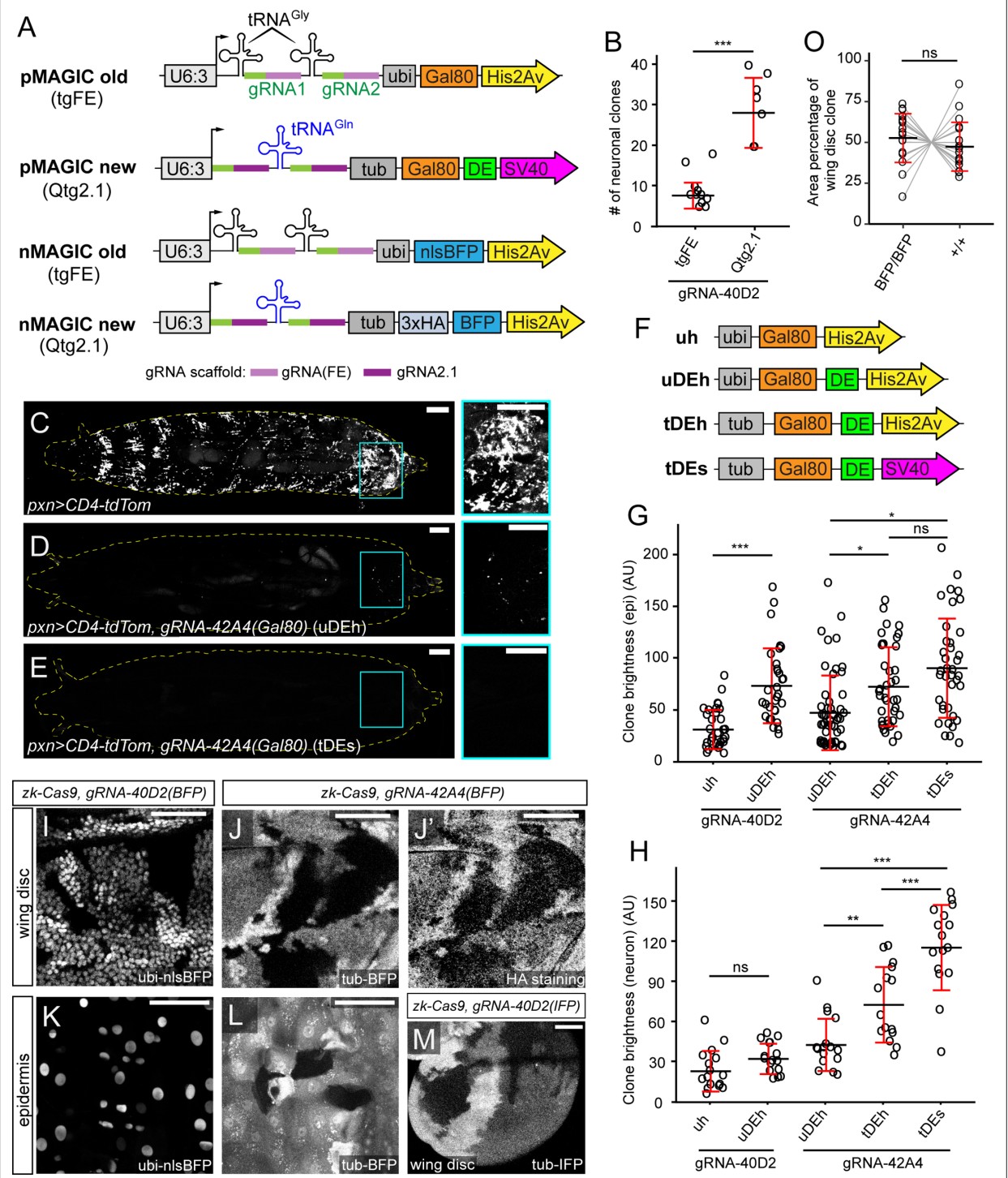

**Figure 1.** New gRNA-marker designs improve pMAGIC and nMAGIC. (**A**) Original and new designs of gRNA-markers for pMAGIC and nMAGIC. (**B**) Comparison of clone frequency in larval sensory neurons between two gRNA designs. Clones were induced by *zk-Cas9* (expressed in the embryonic ectoderm) and labeled by the pan-neuronal driver *RabX4-Gal4>MApHS* (MApHS: pHluorin-CD4-tdTomato). The number represents clones between A1 and A7 segments on one side of each larva. n=larvae number: tgFE (n=10), Qtg2.1 (n=10). (**C–E**) Labeling of hemocytes in whole 3rd instar larvae by *pxn-Gal4>CD4-tdTom* alone (**C**) or together with *gRNA-42A4(Gal80)-uDEH* (*ubi-Gal80*) (**D**) or *gRNA-42A4(Gal80)-tDES* (*tub-Gal80*) (**E**). The panels on the right show enlarged views of the boxed regions. (**F**) Designs of Gal80 variants tested in pMAGIC gRNA-markers. (**G**) The brightness of epidermal clones induced by *zk-Cas9* and labeled by the epidermal driver *R38F11-Gal4>tdTom* in the presence of pMAGIC gRNA-markers. n=image numbers: gRNA-40D2-uH (n=32), gRNA-40D2-uDEH (n=31), gRNA-42A4-uDEH (n=52), gRNA-42A4-tDEH (n=39), gRNA-42A4-tDES (n=38). (**H**) The brightness of neuronal clones induced by *zk-Cas9* and labeled by *RabX4-Gal4>MApHS* in the presence of pMAGIC gRNA-markers. The brightness of tdTom was measured

*Figure 1 continued*

and compared. n=neuron numbers: gRNA-40D2-uH (n=16), gRNA-40D2-uDEH (n=16), gRNA-42A4-uDEH (n=16), gRNA-42A4-tDEH (n=15), gRNA-42A4-tDES (n=16). (**I**) A portion of a larval wing disc containing nMAGIC clones visualized by nlsBFP. (**J** and **J'**) A portion of a wing disc containing nMAGIC clones labeled by cytosolic BFP (**J**) and HA staining (**J'**). (**K**) Epidermal clones on the larva body wall labeled by nlsBFP. (**L**) Epidermal clones visualized by cytosolic BFP. (**M**) A portion of a wing disc containing nMAGIC clones labeled by cytosolic miRFP680 (IFP). (**O**) Sizes of nMAGIC *BFP/BFP* clones and wild-type (*+/+*) clones in wing discs. Two types of clones in the same discs were connected. n=wing disc number: BFP/BFP (n=18), +/+ (n=18). In all plots, black bar, mean; red bar, SD; AU, arbitrary unit. Student's t-test in (**B**); one-way analysis of variance (ANOVA) and Tukey's honest significant difference (HSD) test in (**G**) and (**H**). paired t-test in (**O**) *p≤0.05, **p≤0.01, ***p≤0.001, ns, not significant. For (**C–E**), scale bar, 300 μm. For (**I–M**), scale bar, 100 μm.

The online version of this article includes the following figure supplement(s) for figure 1:

**Figure supplement 1.** MAGIC principles.

labeling of homozygous cells that lose the gRNA-marker. In contrast, nMAGIC uses ubiquitously expressed BFP, which results in brighter labeling of gRNA-marker homozygous cells, intermediate labeling of gRNA-marker heterozygous cells, and lack of labeling of gRNA-marker-negative homozygous cells (***Figure 1—figure supplement 1B***).

We have previously developed gRNA-marker vectors for both pMAGIC and nMAGIC. However, gRNA-markers made with these vectors exhibit some limitations. First, the clone frequency can be low for certain gRNAs (***Allen et al., 2021***). Second, the *ubi-Gal80* in pMAGIC gRNA-markers does not completely suppress Gal4 activity in certain tissues, such as hemocytes. Third, pMAGIC clones are sometimes too dim to visualize cell morphology, such as thin dendrite and axon projections of neurons. Lastly, nMAGIC utilizes a nuclear BFP marker (nlsBFP), which does not show the cell shape and sometimes cannot mark clones effectively. Thus, to optimize MAGIC, we first sought to improve the gRNA-marker designs.

Since clone induction in MAGIC depends on gRNA-induced DNA DSBs, we asked whether a more efficient gRNA design enhances clone frequency in somatic tissues. The previous gRNA-marker vectors used a tgFE design, which contains a flip of A-U positions and a stem-loop extension (*F*+E) of the original gRNA scaffold and a tRNA$^{Gly}$ before each gRNA targeting sequence (***Figure 1A***; ***Allen et al., 2021***). However, we have shown that an improved Qtg2.1 design, which contains an additional extension of the second stem loop in the gRNA scaffold (gRNA2.1) and a single tRNA$^{Gln}$ spacer between the two gRNAs (***Figure 1A***), is much more mutagenic than tgFE in somatic tissues (***Koreman et al., 2021***). We thus compared two pMAGIC gRNA markers that are based on these two gRNA designs but target the same genomic sequences at cytological band 40D2 to assess their ability to induce clones in peripheral sensory neurons. When used with the same neuronal/epidermal precursor Cas9, *zk-Cas9* (***Allen et al., 2021***), Qtg2.1 resulted in three times more neuronal clones as compared to tgFE (***Figure 1B***), confirming a positive correlation between gRNA efficiency and clone frequency.

To ensure more complete suppression of Gal4 activity by Gal80 in pMAGIC, we replaced the *ubi* enhancer driving Gal80 expression with an *αTub84B* (*tub*) enhancer that was used in *tub-Gal80* in the MARCM system (***Lee and Luo, 1999***). When combined with the larval hemocyte marker *pxn-Gal4 UAS-CD4-tdTom* (***Han et al., 2014***; ***Figure 1C***), *ubi-Gal80* did not completely suppress the labeling of hemocytes (***Figure 1D***). In contrast, no labeled hemocytes could be detected in the presence of *tub-Gal80* (***Figure 1E***), suggesting that *tub-Gal80* is a better marker for pMAGIC.

To improve clone brightness in pMAGIC, we sought to destabilize Gal80 and reduce its expression, reasoning that dim clones are due to prolonged Gal80 activity after clone induction. In addition to replacing the *ubi* enhancer with the *tub* enhancer, we also introduced protein and mRNA destabilization sequences (DE) (***Li et al., 1998***; ***Zubiaga et al., 1995***) at the 3' end of the Gal80 coding sequence, and replaced the *His2Av* polyA by *SV40* polyA (***Figure 1F***), as the latter reduces transgene expression (***Han et al., 2011***). By measuring the brightness of epidermal (***Figure 1G***) and neuronal (***Figure 1H***) clones induced by *gRNA-40D2* and *gRNA-42A4*, we found that each of the three changes improved clone brightness.

Finally, to visualize cell shape in nMAGIC, we replaced nlsBFP with cytosolic BFP tagged by 3 X Hemagglutinin (HA), in addition to utilizing the *tub* enhancer. This new design allowed us to better discern clone shapes in both the wing imaginal disc and the epidermis (***Figure 1J, J', and L***), in contrast to the previous *ubi-nlsBFP* design (***Figure 1I and K***). To make nMAGIC compatible with more fluorescent reagents, we generated an additional vector that contains miRFP680, a far-red/infrared

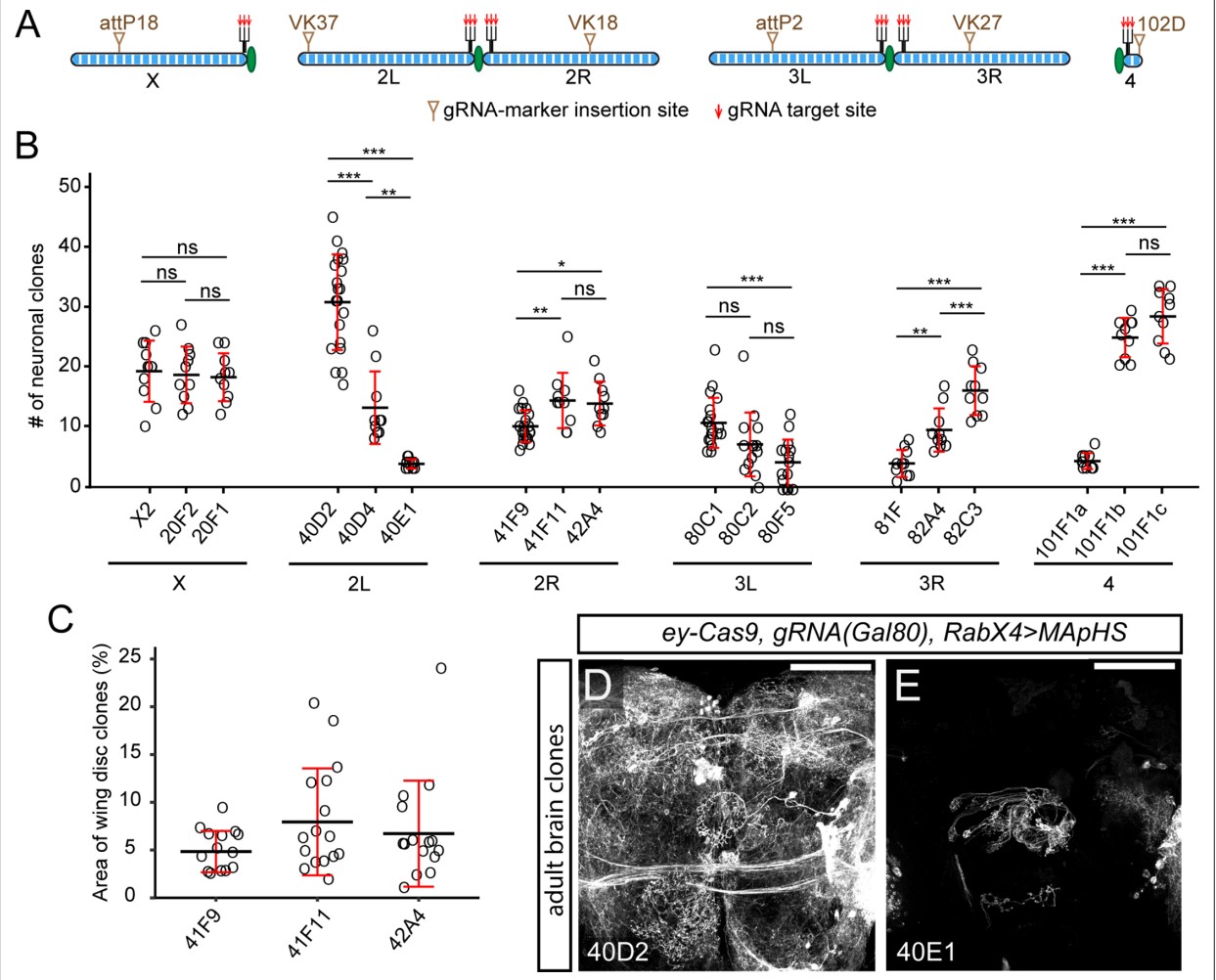

**Figure 2.** A genome-wide gRNA-marker kit suits diverse needs of clone frequency. (**A**) Scheme of gRNA-marker insertion sites and target sites on *Drosophila* chromosomes. (**B**) Comparison of clone frequencies of all pMAGIC gRNA-markers in larval sensory neurons, clones are labeled using *RabX4-Gal4>MApHs* (for Chromosome X, II, and IV) or *21–7 Gal4 UAS-MApHS* (for Chromosome III). n=larvae number: X2 (n=10), 20F2 (n=10), 20F1 (n=10), 40D2 (n=20), 40D4 (n=10), 40E1 (n=10), 41F9 (n=20), 41F11 (n=10), 42A4 (n=10), 80C1 (n=20), 80C2 (n=14), 80F5 (n=15), 81 F (n=10), 82A4 (n=10), 82C3 (n=10), 101F1a (n=10), 101F1b (n=10), 101F1c (n=10). (**C**) Comparison of clone areas in larval wing discs labeled by nMAGIC gRNA markers on 2 R. n=wing disc number: 41F9 (n=14), 41F11 (n=16), 42A4 (n=15). (**D and E**) Neuronal clones in the central part of the adult brain induced by *ey-Cas9* (expressed in progenitor cells of many neuronal tissues) and labeled by *RabX4-Gal4>MApHS* along with pMAGIC gRNA-markers *gRNA-40D2* (**D**) and *gRNA-40E1* (**E**). MApHS contains pHluorin and tdTom (***Han et al., 2014***), but only the tdTom channel is shown. In all plots, black bar, mean; red bar, SD. One-way ANOVA and Tukey's HSD test. *p≤0.05, **p≤0.01, ***p≤0.001, ns, not significant. For (**D**) and (**E**), scale bar 100 μm.

fluorescent protein (IFP) (***Matlashov et al., 2020***), in place of BFP. Wing-disc clones labeled with this marker were readily detectable in unstained tissues (***Figure 1M***). By measuring the sizes of homozygous gRNA-marker clones (*BFP/BFP*) and homozygous wild-type (WT) clones (*+/+*) in wing discs, we found that these two cell populations in the twin spots showed no noticeable bias in growth or viability (***Figure 1O***).

Thus, by altering the designs of the gRNAs and the Gal80 and BFP markers, we created new vectors optimized for more robust applications of nMAGIC and pMAGIC.

## A gRNA-marker kit is established for all four chromosomes of *Drosophila*

To enhance the utility of MAGIC in *Drosophila*, we generated complete sets of pMAGIC and nMAGIC (BFP version) gRNA-markers for all four chromosomes (***Figure 2A***). To identify suitable gRNA target sites, we analyzed the pericentromeric sequences of X, 2 L, 2 R, 3 L, 3 R, and 4, based on three criteria

**Table 1.** gRNA-marker collection.

| Chr Arm | Target site | gRNA location | pMAGIC vector | nMAGIC vector | Clone frequency | BDSC IDs ‡ |
|---|---|---|---|---|---|---|
| 2L | 40D2 | attP[VK00037] | pAC-U63-gRNA2.1-ubiGal80(DE)-His2Av | pAC-U63-tgRNA-nlsBFP* pAC-U63-gRNA2.1-tub-miRFP680-T2A-HO1(HA) (HA) | 31 | 606005 and 92741 |
| 2L | 40D4 | attP[VK00037] | pAC-U63-tgRNA-Gal80* | pAC-U63-tgRNA-nlsBFP* | 13 | 92744 and 92742 |
| 2L | 40E1 | attP[VK00037] | pAC-U63-tgRNA-Gal80* | pAC-U63-tgRNA-nlsBFP* | 4 | 606004 and 606003 |
| 2R | 41F9 | attP[VK00018] | pAC-U63-gRNA2.1-ubiGal80(DE)-His2Av | pAC-U63-gRNA2.1-tubBFP(HA) | 10 | 606006 and 606010 |
| 2R | 41F11 | attP[VK00018] | pAC-U63-gRNA2.1-ubiGal80(DE)-His2Av | pAC-U63-gRNA2.1-tubBFP(HA) | 14 | 606007 and 606009 |
| 2R | 42A4 | attP[VK00018] | pAC-U63-gRNA2.1-tubGal80(DE)-SV40 | pAC-U63-gRNA2.1-tubBFP(HA) | 14 | 606008 and 606011 |
| 3L | 80F5 | attP2 | pAC-U63-gRNA2.1-tubGal80(DE)-His2Av | pAC-U63-gRNA2.1-tubBFP(HA) | 11 | 606014 and 606017 |
| 3L | 80C2 | attP2 | pAC-U63-gRNA2.1-tubGal80(DE)-His2Av | pAC-U63-gRNA2.1-tubBFP(HA) | 7 | 606013 and 606016 |
| 3L | 80C1 | attP2 | pAC-U63-gRNA2.1-tubGal80(DE)-His2Av | pAC-U63-gRNA2.1-tubBFP(HA) | 4 | 606012 and 606015 |
| 3R | 81 F | attP[VK00027] | pAC-U63-gRNA2.1-tubGal80(DE)-His2Av | pAC-U63-gRNA2.1-tubBFP(HA) | 4 | 606020 and 606021 |
| 3R | 82A4 | attP[VK00027] | pAC-U63-gRNA2.1-tubGal80(DE)-His2Av | pAC-U63-gRNA2.1-tubBFP(HA) | 10 | 606018 and 606022 |
| 3R | 82C3 | attP[VK00027] | pAC-U63-gRNA2.1-tubGal80(DE)-His2Av | pAC-U63-gRNA2.1-tubBFP(HA) | 16 | 606019 and 606023 |
| X | X2 | attP18 | pAC-U63-gRNA2.1-tubGal80(DE)-SV40 | pAC-U63-gRNA2.1-tubBFP(HA) | 19 | 606024 and 606028 |
| X | 20F1 | attP18 | pAC-U63-gRNA2.1-tubGal80(DE)-SV40 | pAC-U63-gRNA2.1-tubBFP(HA) | 18 | 606025 and 606779 |
| X | 20F2 | attP18 | pAC-U63-gRNA2.1-tubGal80(DE)-SV40 | pAC-U63-gRNA2.1-tubBFP(HA) | 19 | 606026 and 606029 |
| IV | 101F1-a | attP[ZH-102D]† | pAC-U63-gRNA2.1-tubGal80(DE)-SV40 | pAC-U63-gRNA2.1-tubBFP(HA) | 4 | 606776 and 606780 |
| IV | 101F1-b | attP[ZH-102D]† | pAC-U63-gRNA2.1-tubGal80(DE)-SV40 | pAC-U63-gRNA2.1-tubBFP(HA) | 25 | 606777 and 606781 |
| IV | 101F1-c | attP[ZH-102D]† | pAC-U63-gRNA2.1-tubGal80(DE)-SV40 | pAC-U63-gRNA2.1-tubBFP(HA) | 28 | 606778 and 606782 |

Clone frequencies are measured by averaging clone numbers in 10 laterally mounted 3rd instar larvae with pMAGIC gRNA-markers. Only clones in A1 to A7 segments of one side of each larva were counted.

*Previously published gRNA-markers.

†3xP3-RFP has been removed from these lines.

‡Detailed information for each line is available at https://bdsc.indiana.edu/stocks/misc/magic.html.

(*Allen et al., 2021*): (1) conserved in closely related *Drosophila* species to minimize the chance of single nucleotide polymorphism, (2) located away from functionally critical regions to avoid disrupting essential processes, and (3) unique within the genome to minimize off-target effects. For each MAGIC construct, we selected a pair of non-repetitive gRNA target sequences in intergenic regions to maximize the chances of DSBs. These two sequences are closely linked to minimize the risk of large deletions. Given the variable efficiency of gRNA target sequences, we selected three pairs of gRNAs targeting three chromosomal locations for each chromosomal arm and named them according to the corresponding cytoband (*Table 1*).

To evaluate the clone-induction properties of these gRNAs, we combined the pMAGIC set with *zk-Cas9* and counted the number of neuronal clones in A1–A7 larval hemi-segments (*Figure 2B*). As expected, the clone frequency varied from gRNA to gRNA, but we were able to identify efficient gRNAs (≥ 10 clones per larva) for every chromosomal arm. We previously found that different gRNAs follow the same trend of relative efficiency in different tissues (*Allen et al., 2021*). Here we additionally tested nMAGIC gRNAs for 2 R in the wing disc (*Figure 2C*) and noticed a similar trend of clone induction to that of their pMAGIC counterparts in sensory neurons (*Figure 2B*), suggesting that the results in sensory neurons are transferrable to other tissues.

Certain gRNAs (e.g. gRNA-40E1) exhibited very low clone frequency. Such gRNAs can be useful for inducing sparse clones in highly packed tissues such as the brain. For example, using the same *ey-Cas9* (*Ji et al., 2022*), the highly efficient gRNA-40D2 induced too many neuronal clones in the adult brain for morphological analysis (*Figure 2D*), while gRNA-40E1 gave rise to few clones, whose projection patterns were much easier to analyze (*Figure 2E*).

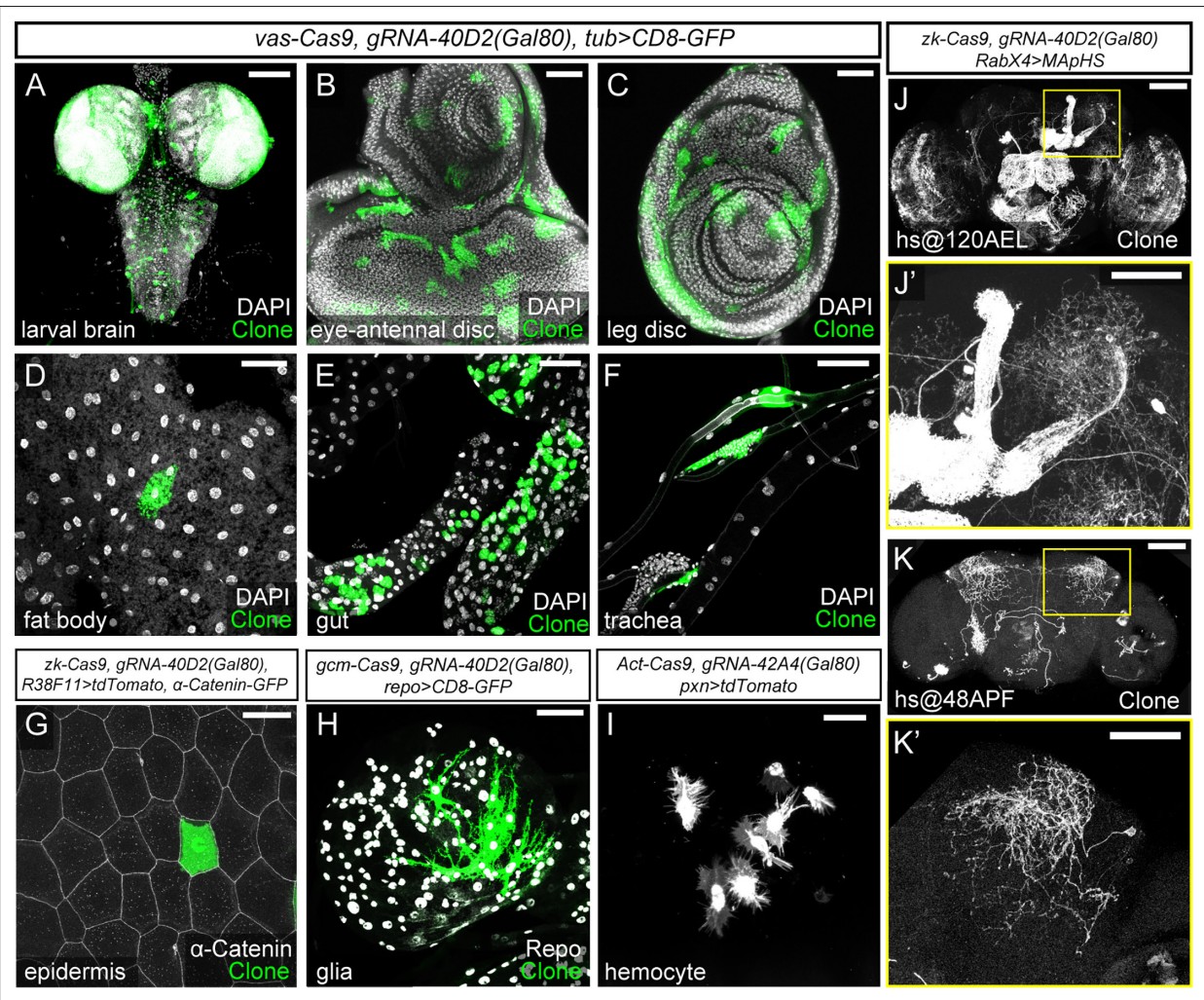

**Figure 3.** MAGIC allows clonal analysis in diverse tissues and cell types. (**A–F**) pMAGIC clones induced in different tissues by *vas-Cas9* (ubiquitous Cas9) *gRNA-40D2(Gal80)* and labeled by *tub-Gal4 UAS-CD8-GFP* (green). DAPI staining (white) shows all nuclei. (**G**) A pMAGIC epidermal clone on the larval body wall induced by *zk-Cas9 gRNA-40D2(Gal80)* and labeled by *R38F11-Gal4>tdTom* (green). Epidermal junctions are labeled by α-Catenin-GFP (white). (**H**) pMAGIC glia clones in the larval brain induced by *gcm-Cas9* (expressed in glial precursor genes) *gRNA-40D2(Gal80)* and labeled by *repo-Gal4 UAS-CD8-GFP* (green). Glial nuclei are labeled by Repo staining (white). (**I**) pMAGIC hemocyte clones induced by *Act-Cas9 gRNA-40D2(Gal80)* and labeled by *pxn-Gal4>CD4-tdTom*. (**J-K'**) pMAGIC clones in adult brain induced by *hs-Cas9 gRNA-40D2(Gal80)* and labeled by *RabX4-Gal4>MApHS*. Heat shock was performed at 120 hr after egg lay (AEL) (**J-J'**) and 48 hr after puparium formation (APF) (**K-K'**). The boxed areas were enlarged to show clones in the mushroom body and lateral horn region. Only the tdTom channel is shown. In (**A**), (**D–F**), (**H**), (**J**), and (**K**), scale bar 100 μm; in (**B–C**), (**G**), (**J'**), and (**K'**), scale bar 50 μm; in (**I**), scale bar 25 μm.

Together, the pMAGIC and nMAGIC gRNA-marker lines constitute a complete kit (*Table 1*) for genome-wide MAGIC applications in *Drosophila*.

## MAGIC allows clonal analysis in diverse tissues and cell types

To determine if MAGIC can be applied to diverse tissues in *Drosophila*, we conducted clonal analysis in the larva using *gRNA-40D2(Gal80)* and several tissue-specific Cas9s. With the ubiquitous *vas-cas9* (*López Del Amo et al., 2022*) and *tub-Gal4 >UAS-mCD8-GFP*, we readily detected clones in the larval brain (*Figure 3A*), proliferating tissues like eye and leg discs (*Figure 3B–C*), and polyploid tissues like the fat body, gut, and trachea (*Figure 3D–F*). Clone induction in polyploid tissues suggests that crossing-over events occurred before the last cell division. Using *zk-Cas9* and *R38F11-Gal4>UAS-tdTomato (tdTom)*, we observed frequent epidermal clones (*Figure 3G*). Using the glia precursor *gcm-Cas9* and *repo-Gal4 >UAS-mCD8-GFP*, we detected individual glial clones in the brain (*Figure 3H*). The new pMAGIC gRNA-marker design allowed us to reliably induce hemocyte clones (*Figure 3I*).

The ability to control the timing of clone induction has been instrumental for neuronal birth dating and modulation of clone frequency in traditional MARCM analysis of the *Drosophila* adult brain. To explore the potential of pMAGIC to serve similar purposes, we used heat-shock (hs) Cas9 (*Garcia-Marques et al., 2020*), along with *gRNA-40D(Gal80)* and a pan-neuronal Gal4/UAS-membrane marker combination (*RabX4-Gal4>UAS-MApHS*), to induce neuronal clones in the adult brain. Heat shock at 120 hr after egg laying (AEL) induced too many clones for separating individual neurons (*Figure 3J–J'*), while later heat shock at 48 hr after pupal formation (APF) produced many fewer, spatially separated clones in the central brain. These results collectively demonstrate MAGIC's efficacy and flexibility in generating clones in diverse *Drosophila* tissues, indicating its value in studying gene function and cell lineage in various tissue contexts.

The neuromuscular junction (NMJ) in *Drosophila* larvae has been a valuable model for elucidating synaptic biology, owing to its simplicity, accessibility, and conserved features shared with mammalian excitatory synapses (*Charng et al., 2014*; *Menon et al., 2013*; *Chou et al., 2020*). Despite its popularity, the NMJ has rarely been analyzed by the MARCM technique in the literature, likely due to the difficulty of inducing clones in motor neurons. To test the effectiveness of MAGIC in analyzing gene function at the NMJ, we selected two genes crucial for synaptic function and construction: *Vesicular glutamate transporter* (*VGlut*) (*Daniels et al., 2004*) and *bruchpilot* (*brp*) (*Kittel et al., 2006*; *Wagh*

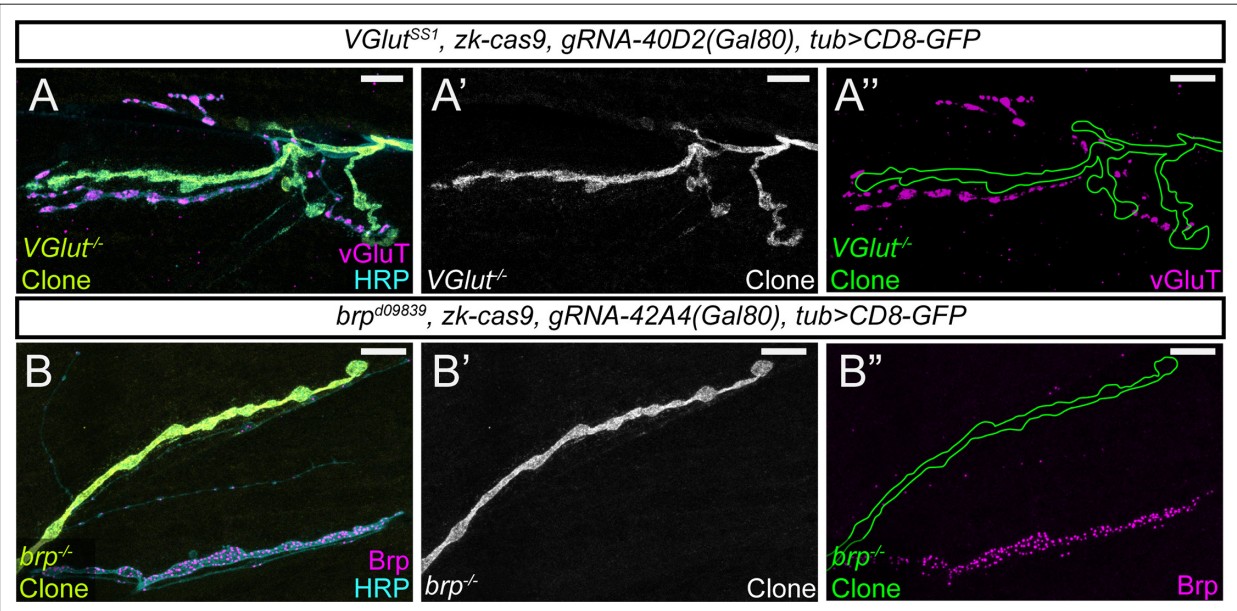

**Figure 4.** MAGIC facilitates clonal analysis at the NMJ. (**A-A"**) pMAGIC clones of *VGlut¹* mutation in motor neurons at the neuromuscular junction. Clones were induced by *zk-Cas9 gRNA-40D2(Gal80)* and labeled by *tub-Gal4>CD8* GFP. The loss of VGlut is confirmed by VGlut staining. The mutant clones are outlined in (**A"**). (**B-B"**) A pMAGIC clone of *brp^d09839* mutation in a motor neuron at the neuromuscular junction. Clones were induced by *zk-Cas9 gRNA-42A4(Gal80)* and labeled by *tub-Gal4>CD8* GFP. The loss of Brp is confirmed by Brp staining. The mutant clone is outlined in (**B"**). In both experiments, HRP staining shows all axons. Scale bars, 10 μm.

*et al., 2006*), null mutations of each of which exhibit recessive lethality in larvae. Combined with appropriate gRNA(Gal80) lines (42A4 for *VGlut* on 2 R and 40D2 for *brp* on 2 L) and *tub-Gal4 UAS-CD8-GFP*, *zk-Cas9* induced frequent clones in type Ib boutons (*Figure 4A–B"*). The loss of VGlut or Brp specifically in GFP-labeled NMJs was confirmed by immunostaining. These results exemplify the value of pMAGIC for dissecting gene function in NMJ biology at the single-cell level.

## MAGIC enables clonal analysis of pericentromeric genes, 4th chromosome-associated mutations, and in interspecific hybrid animals

Because all FRT sites in the existing FRT/Flp mosaic system are located at some distances away from centromeres, it has not been possible to study genes located between the FRT sites and the corresponding centromeres by clonal analysis. In contrast, the gRNA target site in a MAGIC experiment can be user-selected to enable clonal analysis of any given gene. To illustrate this potential, we used *gRNA-41F9* to induce homozygous mutant clones of *Ecdysone receptor* (*EcR*), which is located at 42A10 and is inaccessible to all FRT sites on 2 R. EcR is a transcription factor required for neuronal remodeling during metamorphosis (*Brown et al., 2006*). In peripheral sensory neurons, EcR activation at the beginning of pupariation causes apoptosis of some dendritic arborization (da) neurons

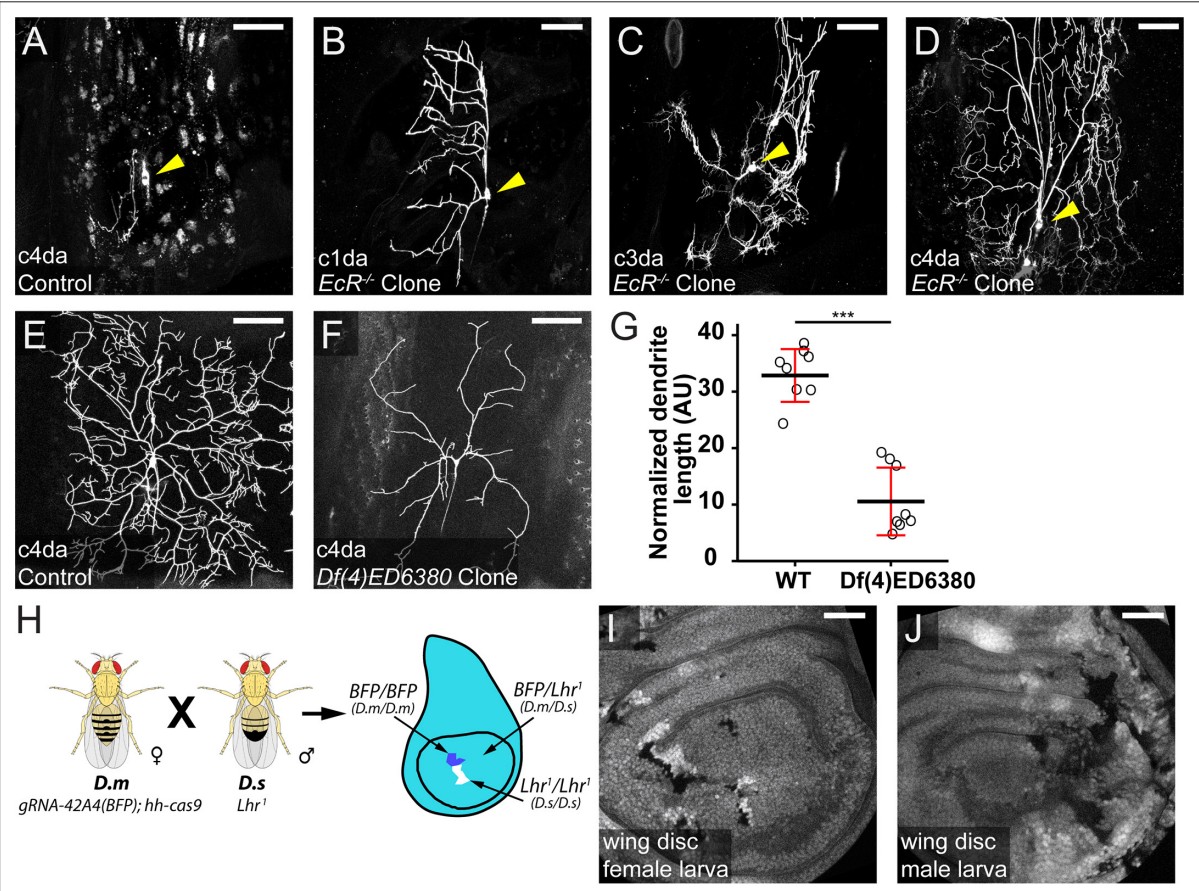

**Figure 5.** MAGIC enables clonal analysis of pericentromeric genes, 4th chromosome-associated mutations, and in interspecific hybrid animals. (**A**) A WT pMAGIC class IV da neuron clone exhibiting complete dendrite pruning at 16 hr APF. (**B–D**) pMAGIC clones of *EcR^{M554fs}* mutation in da neurons imaged at 16 hr APF, exhibiting the lack of pruning (**B** and **D**) or apoptosis (**C**). In (**A–D**), the clones were induced by *zk-cas9* with *gRNA-41F9(Gal80)* and labeled by *RabX4-Gal4>MApHS*. Neuronal cell bodies are indicated by arrows. Only the tdTom channel is shown. The signals in epidermal cells (**A**) were due to engulfment of pruned dendrites by epidermal cells (*Han et al., 2014*). (**E** and **F**) WT (**E**) and *Df(4)ED6380* (**F**) pMAGIC clones in C4da neurons induced by *zk-cas9 gRNA-101Fc(Gal80)* and labeled by *RabX4-Gal4>MApHS*. Only the tdTom channel is shown. (**G**) Normalized dendrite length of WT clones and deficiency clones. Black bar, mean; red bar, SD. Student's t-test. ***≤0.001. (**H**) Scheme for interspecific crosses between *D. melanogaster* (*D.m*) and *D. simulans* (*D.s*). (**I** and **J**) Wing discs from male (**I**) and female (**J**) progeny carrying clones. Scale bars, 50 μm.

The online version of this article includes the following figure supplement(s) for figure 5:

**Figure supplement 1.** Transgenic markers on the 4th chromosome show uneven expression.

while triggering dendrite pruning of other da neurons (*Williams and Truman, 2005*). As expected, a wild-type (WT) pMAGIC clone of class IV da (C4da) neuron exhibited complete dendrite pruning at 16 hr after puparium formation (APF), accompanied by dendrite debris phagocytosed into epidermal cells (*Han et al., 2014*; *Figure 5A*). In contrast, *EcR* mutant da neuron clones still maintained larval dendritic arbors at 16 hr APF (*Figure 5B–D*), instead of dying (in the case of C3da) or undergoing dendrite pruning (in the cases of C1da and C4da).

Mosaic analysis of genes located on the 4th chromosome had not been possible until the recent introduction of FRT sites onto this chromosome by CRISPR-mediated knock-in (*Goldsmith et al., 2022*). Despite these advances, existing mutations on FRT-lacking 4th chromosomes still cannot be analyzed by the FRT/Flp system, given that meiotic recombination is exceedingly rare on the 4th chromosome, preventing introduction of FRT sites onto mutant chromosomes. A valuable gene-disruption resource in *Drosophila* is the deficiency library consisting of strains harboring chromosomal deletions. MAGIC can potentially be used in conjunction with the deficiency library to screen for genes important in a particular biological process. To test the potential of MAGIC for analyzing mutations on the 4th chromosome and for gene discovery with deficiencies, we generated pMAGIC C4da clones of *Df(4)ED6380*, a deficiency that deletes a segment between cytological bands 102B7 and 102D5. These clones show dramatically reduced dendrites compared to the WT controls (*Figure 5E–G*), indicating the existence of genes important for dendrite growth in this region.

When examining nMAGIC gRNA-markers for the 4th chromosome, we noticed uneven expression of *tub-3xHA-BFP* in epidermal cells and some imaginal tissues, exemplified by cells lacking detectable BFP signal (*Figure 5—figure supplement 1A* and B). This variegated expression is likely due to transgenes residing in heterochromatin and is common for transgenes located on the 4th chromosome (*Riddle et al., 2011*). While this uneven expression limits the usefulness of our gRNA-markers in the corresponding epithelial tissues, Gal80 in pMAGIC gRNA-markers efficiently suppressed Gal4 activities in most neurons (*Figure 5—figure supplement 1C*), confirming their effectiveness in neuronal MAGIC analysis.

Lastly, we wondered if the MAGIC system can generate clones in interspecific hybrid animals derived from *D. melanogaster* and *D. simulans* parents, given that the two species show a large degree of synteny (*Chakraborty et al., 2021*). The clones in these animals would contain homozygous chromosomal arms derived from a single species. To test this idea, we crossed *D. melanogaster* females containing *gRNA-42A4(BFP); hh-cas9* to *D. simulans* males carrying a loss of function mutation of *Lethal hybrid rescue mutation* (*Lhr*), which ensures the viability of the hybrid male progeny (*Barbash, 2010*). Indeed, we observed twin spots consisting of dark clones, containing only *D. simulans* 2 R, and brighter clones, containing only *D. melanogaster* 2 R, in wing discs of both female and male progeny (*Figure 5G and H*), demonstrating the feasibility of studying species-specific alleles in cell-cell interactions in interspecific hybrids.

## Discussion

Conceptually, MAGIC is a simpler and more convenient mosaic technique compared to traditional recombinase-dependent methods. Theoretically, it can be applied directly to any existing stock, including those that are not currently compatible with the FRT/Flp system. However, the lack of gRNA-marker transgenes has prevented its wide application in *Drosophila*. In this study, we present a complete gRNA-marker kit that enables genome-wide MAGIC in *Drosophila*, removing this bottleneck. The new gRNA markers incorporate optimized designs that improve clone frequency and labeling in both pMAGIC and nMAGIC. We further demonstrate the compatibility of MAGIC with broad tissues and cell types. More importantly, MAGIC enables mosaic analyses that could not be accomplished by existing FRT/Flp systems, such as those of pericentromeric genes, deficiency chromosomes, and interspecific homologous chromosomes. Thus, this MAGIC kit provides *Drosophila* researchers with greater flexibility for conducting mosaic analyses of diverse purposes.

To implement MAGIC, one needs to first choose a proper Cas9 that is expressed in the precursor cells of the targeted cell population. We show that ubiquitously expressed Cas9 lines, such as *vas-Cas9* (*López Del Amo et al., 2022*) and *Act5C-Cas9* (*Port et al., 2014*), are sufficient to induce clones in broad tissues. Alternatively, heat shock (HS)-induced Cas9 (*Garcia-Marques et al., 2020*) can offer temporal control of clone induction in most tissues, akin to the HS-Flp commonly used in FRT-based mosaic analysis. The third option is a Cas9 driven by a tissue-specific enhancer specifically

in the precursor cells of the target tissues. Examples shown in this study include *ey-Cas9* expressed in many neuronal lineages (*Ji et al., 2022*), *gcm-Cas9* expressed in glial progenitor cells (*Chen et al., 2024*), *zk-Cas9* expressed in precursor cells of epidermal cells, motor neurons, and somatosensory neurons (*Koreman et al., 2021*), and *hh-Cas9* expressed in imaginal tissues (*Poe et al., 2019*). These Cas9 lines have the advantages of being more specific and efficient, and more convenient to use than ubiquitous or inducible ones. Multiple strategies, including enhancer fusion (*Port et al., 2014*; *Poe et al., 2019*), Gal4-to-Cas9 conversion (*Koreman et al., 2021*; *Chen et al., 2020*), and insertion of Cas9 in specific gene loci (*Chen et al., 2024*), have been developed to ease the generation of such tissue-specific Cas9 lines. As an ongoing effort, we have been converting Gal4 lines known to be expressed in progenitor cells into Cas9 (available here), in the hope of making MAGIC accessible to more *Drosophila* tissues. It is worth noting that many Cas9 lines show leaky activity in the germline, which could mutate and inactivate the target sequence in the presence of a gRNA. Thus, it is not recommended to combine Cas9 and gRNA transgenes in the same strain as a long-term stock, unless a Cas9 inhibitor can also be introduced into this stock to suppress the germline activity.

The second component of MAGIC is a gRNA-marker line that resides on the appropriate chromosome arm and targets Cas9 to cut a pericentromeric site on the same arm. In the gRNA-marker kit, we generated three lines targeting different sites for each chromosome arm. These lines exhibit a broad range of clone frequencies in larval sensory neurons. As different gRNA markers maintain similar relative efficiencies of clone induction across tissues (*Allen et al., 2021*), one can choose a gRNA marker more appropriate for their applications based on the target gene location and the desired clone frequency. Notably, a higher clone frequency may not always be beneficial, such as when studying the projection patterns of individual neurons in the brain. A low-efficiency gRNA marker in this kit may be more desirable in such applications.

Besides commonly conducted mosaic analysis in *Drosophila*, MAGIC also enables many novel analyses that are difficult or impossible to accomplish with traditional systems. One example is to analyze interactions among species-specific cells in an interspecific hybrid animal for understanding the cell biological basis of hybrid incompatibility. Interspecific crosses between WT *D. melanogaster* and WT *D. simulans* result in sex-specific lethality of F1 progeny (*Barbash, 2010*). The cellular basis of this lethality is poorly understood. With MAGIC, one may generate mixed cell populations that are homozygous or heterozygous for species-specific alleles in the same hybrid embryo or larva, in which the relative fitness of the three cell populations can be compared. Similarly, MAGIC could complement genome-wide association studies (GWAS) using wild-derived isogenic strains and provide much more mechanistic detail. Such strains as those in the *Drosophila* Genetic Reference Panel (DGRP) (*Mackay et al., 2012*) have been very useful for discovering loci that are responsible for phenotypic variations. MAGIC can be combined with these strains to analyze the effects of homozygosity of specific variants at the cell biology level (*Allen et al., 2021*). In addition, MAGIC can be combined with the *Drosophila* deficiency kit for rapid genome-wide mosaic screens. A deficiency exhibiting a desired phenotype in such screens can be further dissected with smaller deficiencies within the deleted region or existing mutations in candidate genes. In this way, the convenience of MAGIC could significantly accelerate phenotype-based gene discovery. Lastly, because the crossover site in MAGIC can be user-defined, one can easily generate gRNA-markers that induce somatic recombination at specific genome locations. With this feature, it is possible to induce crossover between two mutant alleles on the same chromosomal arm to generate clones homozygous for the distal mutation but heterozygous for the proximal mutation. This flexibility could also be useful for mapping undefined genetic loci that are responsible for certain phenotypes, especially in non-traditional model organisms.

## Materials and methods
### Fly stocks and husbandry
See the Key Resource Table for details of fly stocks used in this study. Most fly lines were either generated in the Han lab or obtained from the Bloomington *Drosophila* Stock Center. *lethal hybrid rescue* (*lhr*) mutant *D. simulans* was a gift from Dr. Dan Barbash. All flies were grown on standard yeast-glucose medium, in a 12:12 light/dark cycle, at 25 °C unless otherwise noted. Virgin males and females for mating experiments were aged for 3–5 days.

To generate and label pMAGIC clones in larval peripheral sensory neurons, we used either *RabX4-Gal4 UAS-MApHS* (for gRNA-markers on chromosomes X, II, and IV) or *21–7 Gal4 UAS-MApHS* (for gRNA-markers on chromosome III) combined with *zk-cas9*. To count peripheral sensory neuronal clones on the larval body wall, third instar larvae were mounted laterally on slides and then counted in segment A1-A7 under a Nikon SMZ18 stereomicroscope. pMAGIC clones were induced and labeled by *RabX4-Gal4 UAS-MApHS* combined with *ey-cas9* or *hs-cas9* in the fly adult brain, by *tub-Gal4 UAS-mCD8-GFP* combined with *vas-cas9* in larval brains, imaginal discs, fat bodies, guts, and trachea, by *repo-Gal4 UAS-mCD8-GFP* combined with *gcm-cas9* in glia, by *pxn-Gal4 UAS-tdTom* combined with *Act-cas9* in hemocytes, by *tub-Gal4 UAS-mCD8-GFP* combined with *zk-cas9* in larval motor neurons, by *R38F11-Gal4 UAS-tdTom* combined with *zk-cas9* in the larval epidermis. To induce nMAGIC clones in wing imaginal discs of interspecific hybrid animals, we use *gRNA-42A4(BFP); hh-cas9* virgin females of *D. melanogaster* to cross with *Lhr[1]* (**Brideau et al., 2006**) *D. simulans* males.

## Molecular cloning

### MAGIC gRNA cloning vectors

pMAGIC gRNA-marker vectors constructed in this study include pAC-U63-gRNA2.1-ubiGal80(DE)-His2Av, pAC-U63-gRNA2.1-tubGal80(DE)-His2Av and pAC-U63-gRNA2.1-tubGal80(DE)-SV40. To make pAC-U63-gRNA2.1-ubiGal80(DE)-His2Av, a fragment containing gRNA2.1 scaffold (gRNA2.1) and U6 3' flanking sequence (U63fl) was first used to replace gRNA2.1-QtRNA-Gal80(TS)-gRNA2.1-U63fl in pAC-U63-QtgRNA2.1–8 R (Addgene 170514; **Koreman et al., 2021**). Then a destabilization sequence (DE) containing a GS linker, amino acids (AAs) 422–461 of mouse ornithine decarboxylase (**Zubiaga et al., 1995**), 2 X RNA destabilizing nonamer (TTATTTATTgatccTTATTTATT) (**Zubiaga et al., 1995**) was added to the C-terminus of Gal80 in frame. To make pAC-U63-gRNA2.1-tubGal80(DE)-His2Av, a 2.6 kb *tub* enhancer was amplified from pENTR221-tubP (**Chen et al., 2025**) by PCR and used to replace the *ubi* enhancer in pAC-U63-gRNA2.1-ubiGal80(DE)-His2Av. To make pAC-U63-gRNA2.1-tubGal80(DE)-SV40, a SV40 polyA sequence was amplified from pAPIC-PHCS (**Han et al., 2011**) and used to replace the His2Av polyA in pAC-U63-gRNA2.1-ubiGal80(DE)-His2Av. To make the nMAGIC gRNA-marker vector pAC-U63-gRNA2.1-tubBFP(HA), the gRNA2.1-U63fl fragment was first used to replace gRNA2.1-QtRNA-BFP(TS)-gRNA2.1-U63fl in pAC-U63-QtgRNA2.1-BR (Addgene 170513; **Koreman et al., 2021**). The *tub* enhancer was then used to replace the *ubi* enhancer. Lastly, a synthetic DNA fragment (GenScript) encoding 3 X HA was used to replace the nuclear localization signal at the N-terminus of mTagBFP. To make nMAGIC gRNA-marker vector pAC-U63-gRNA2.1-tub-miRFP680-T2A-HO1(HA), an miRFP680-T2A-HO1 coding sequence was used to replace BFP in pAC-U63-gRNA2.1-tubBFP(HA). HO1 encodes heme oxygenase 1 and is necessary for generating the chromophore of miRFP680. Cloning was carried out by ligation with T4 ligase or NEBuilder DNA Assembly reactions (New England Biolabs Inc).

### MAGIC gRNA expression vectors

34 gRNA expression vectors were constructed with the corresponding gRNA cloning vectors as listed in **Table 1** according to published protocols (**Koreman et al., 2021**). Briefly, for each expression vector, two primers containing appropriate gRNA target sequences **Supplementary file 1** were used to amplify a fragment consisting of 3' end of U6:3 promoter, the first target sequence (TS1), gRNA2.1, tRNA$^Q$, the second target sequence (TS2), and the beginning sequence of gRNA2.1 with pAC-U63-QtgRNA2.1-BR as the PCR template. The PCR product was then assembled with SapI-digested gRNA cloning vectors using NEBuilder DNA Assembly.

Injections were carried out by Rainbow Transgenic Flies (Camarillo, CA 93012 USA) or Genetivision (Stafford, TX 77477) to transform flies through φC31 integrase-mediated integration into attP docker sites. The 3xP3-RFP selection marker in gRNA markers inserted at the attP[ZH-102D] site was removed by crossing to Cre.

## Live imaging

Live imaging of larval epidermal cells, sensory neurons, and hemocytes was performed as previously described (**Poe et al., 2017**). Animals were collected at 96 (for late third larvae) or 120 hr (for wandering third instar larvae) AEL and mounted in glycerol on a slide with vacuum grease as a spacer. Animals were imaged using a Leica SP8 confocal microscope with a 40 X NA1.3 oil objective, pinhole size 2 airy

units, and a z-step size of 1 μm. For the epidermis, images were taken at the dorsal midline of A2 and A3 segments. For dendritic arborization neurons, images were taken from A1 to A7 hemi-segments.

For imaging sensory neurons in pupae, newly formed pupae were collected and incubated at 25 °C. After 16 hr, the pupal cases were carefully removed, and the pupae were mounted dorsal side up on slides with halocarbon oil beneath a coverslip. Double-sided tape was used as a spacer. The mounted pupae were imaged using a Leica SP8 confocal microscope with a 40 X NA1.3 oil objective.

## Heat shock induction of neuronal clones in the adult brain

To induce heat shock, vials containing animals at the appropriate developmental stages were submerged in a 37 °C water bath for 1 hr. Subsequently, the vials were transferred back to a 25 °C incubator until adults emerged.

## Imaging of wing discs and other larval tissues

Larval dissections were performed as described previously (*Poe et al., 2019*). Briefly, wandering third instar larvae were dissected in a small petri dish filled with cold phosphate-buffered saline (PBS). The anterior half of the larva was inverted. To prepare imaginal discs, trachea, and gut were removed. Samples were then transferred to 4% formaldehyde in PBS and fixed for 20 min at room temperature. After washing with PBS, the tissues were stained in DAPI (1:1000) in 0.2% PBST (PBS with 0.2% Triton X-100) for 5 min. The tissues were washed again in PBST and mounted in SlowFade Diamond Antifade Mountant (Thermo Fisher Scientific) on a glass slide. A coverslip was lightly pressed on top. Imaginal discs were imaged using a Leica SP8 confocal microscope with a 20 X NA0.8 oil objective.

## Adult brain imaging

Flies were aged for 1 day after eclosion. Brains were dissected in PBS at room temperature and then fixed in 4% paraformaldehyde in PBS with constant circular rotation for 20 min at room temperature. The brains were subsequently washed in 0.2% PBST and mounted on a glass slide under a glass coverslip. Vacuum grease was used as a spacer between the coverslip and the slide. Brains were imaged using a Leica SP8 confocal microscope with a 40 X NA1.3 oil objective.

## Larval fillet preparation

Larval fillet dissection was performed on a petri dish half-filled with PMDS gel. Wandering third instar larvae were pinned on the dish in PBS dorsal-side up and then dissected to expand the body wall. PBS was then removed, and 4% formaldehyde in PBS was added to fix larvae for 15 min at room temperature. For VGlut staining, the fillets were fixed in Bouin's solution for 5 min at room temperature. Fillets were rinsed and then washed at room temperature in PBS for 20 min or until the yellow color from Bouin's solution faded. After immunostaining, the head and tail of fillets were removed, and the remaining fillets were placed in SlowFade Diamond Antifade Mountant on a glass slide. A coverslip was lightly pressed on top. Larval fillets were imaged using a Leica SP8 confocal microscope with a 40 X NA1.3 oil objective.

## Immunohistochemistry

Larval brains and larval fillets were rinsed and washed at room temperature in 0.2% PBST after fixation. The samples were then blocked in PBST with 5% normal donkey serum (NDS) for 1 hr before incubating with appropriate primary antibodies in the blocking solution. Brains were stained for 2 hr at room temperature, and fillets were stained overnight at 4 °C. After additional rinsing and washing, the samples were incubated with secondary antibodies for 2 hr at room temperature. The samples were then rinsed and washed again before mounting and imaging. Primary antibodies used in this study are mouse anti-Repo antibody 8D12 (1:50 dilution), mouse anti-Brp antibody nc82 (1:100 dilution), rabbit anti-VGlut (1:200 dilution; *Chen et al., 2024*), mouse anti-HA antibody (12CA5, 1:100), and goat anti-HRP conjugated with Cy3 (1:200). Secondary antibodies include donkey anti-mouse antibody conjugated with Cy5 (1:400) and donkey anti-rabbit antibody conjugated with Cy5 (1:400).

## Image analysis and quantification

ImageJ analyses were conducted in Fiji/ImageJ. To compare brightness of clones in sensory neurons and epidermal cells, clones were detected based on thresholds to generate masks. The clone

brightness within the masks was then measured as mean pixel intensity. To quantify clones in wing discs, two to three slices of optical sections in the middle of the disc were projected into a two-dimensional (2D) image. Clones were detected based on a fixed threshold to generate masks, and the total area of clones in the wing pouch of each disc was measured.

The tracing and measurement of the neuron dendrites were done as previously described in detail (*Poe et al., 2017*). Briefly, dendrites were segmented using local thresholding. The segments were then converted into single-pixel-width skeletons. The total length of skeletons was calculated based on pixel distance. Normalized dendrite length was calculated as dendritic length (μm)/segment width (μm).

### Statistical analysis

One-way analysis of variance (ANOVA) with Tukey's honest significant difference (HSD) test was used when the dependent variable was normally distributed and there was approximately equal variance across groups. A Student's t-test or paired t-test was used when two groups were compared. For additional information on the number of samples, see figure legends. R Studio was used for all statistical analyses.

## Acknowledgements

We thank Dion Dickman, Marcus Smolka, and Developmental Studies Hybridoma Bank (DSHB) for antibodies; Dan Barbash, Ethan Bier, and Bloomington Drosophila Stock Center for fly stocks; Dan Barbash and Tianzhu Xiong for advice on interspecific crosses; Claire Ho and Christina Breneman for helping to establish and test transgenes; Dan Barbash and Mariana Wolfner for feedback on the manuscript.

## Additional information

### Funding

| Funder | Grant reference number | Author |
| --- | --- | --- |
| NIH Office of the Director | R24OD031953 | Yifan Shen<br>Ann T Yeung<br>Payton Ditchfield<br>Elizabeth Korn<br>Rhiannon Clements<br>Xinchen Chen<br>Bei Wang<br>Zixian Huang<br>Michael Sheen<br>Parker A Jarman<br>Chun Han |

The funders had no role in study design, data collection and interpretation, or the decision to submit the work for publication.

### Author contributions

Yifan Shen, Conceptualization, Resources, Data curation, Software, Formal analysis, Supervision, Validation, Investigation, Visualization, Methodology, Writing – original draft, Writing – review and editing; Ann T Yeung, Conceptualization, Data curation, Software, Formal analysis, Supervision, Investigation, Visualization, Methodology; Payton Ditchfield, Bei Wang, Resources; Elizabeth Korn, Data curation, Supervision, Investigation; Rhiannon Clements, Data curation, Formal analysis, Supervision, Investigation, Visualization; Xinchen Chen, Zixian Huang, Michael Sheen, Investigation; Parker A Jarman, Data curation, Investigation; Chun Han, Conceptualization, Supervision, Funding acquisition, Methodology, Writing – original draft, Writing – review and editing

### Author ORCIDs

Yifan Shen ![ORCID] https://orcid.org/0000-0002-3921-1011
Bei Wang ![ORCID] https://orcid.org/0000-0003-3002-3302

Chun Han ⓘ https://orcid.org/0000-0001-7319-8095

Reviewer #1 (Public review): https://doi.org/10.7554/eLife.108453.3.sa1
Reviewer #2 (Public review): https://doi.org/10.7554/eLife.108453.3.sa2
Reviewer #3 (Public review): https://doi.org/10.7554/eLife.108453.3.sa3
Author response https://doi.org/10.7554/eLife.108453.3.sa4

---

## Additional files

### Supplementary files

MDAR checklist

Supplementary file 1. gRNA target sequences.

### Data availability

All data generated or analyzed during this study are included in the manuscript and supporting files. The MAGIC gRNA-marker fly stocks generated in this study have been deposited to the Bloomington Drosophila Stock Center. Plasmids have been deposited to Addgene.

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

# Appendix 1

## Appendix 1—key resources table

| Reagent type (species) or resource | Designation | Source or reference | Identifiers | Additional information |
|---|---|---|---|---|
| Genetic reagent (*D. melanogaster*) | gRNA-40D2-tgFE-uH | **Allen et al., 2021** | | *w; gRNA-40D2-tgFE-uH*$^{VK00037}$ |
| Genetic reagent (*D. melanogaster*) | gRNA-40D2-Qtg2.1-uDEH | This study | RRID:BDSC_606005 | *w; gRNA-40D2-Qtg2.1-uDEH*$^{VK00037}$; in Materials and methods |
| Genetic reagent (*D. melanogaster*) | gRNA-40D4-tgFE-uH | **Allen et al., 2021** | | *w; gRNA-40D4-tgFE-uH*$^{VK00037}$ |
| Genetic reagent (*D. melanogaster*) | gRNA-40E1-tgFE-uH | **Allen et al., 2021** | RRID:BDSC_606004 | *w; gRNA-40E1-tgFE-uH*$^{VK00037}$ |
| Genetic reagent (*D. melanogaster*) | gRNA-42A4-Qtg2.1-uDEH | This study | | *w; gRNA-42A4-Qtg2.1-uDEH*$^{VK00018}$; in Materials and methods |
| Genetic reagent (*D. melanogaster*) | gRNA-41F9-Qtg2.1-uDEH | This study | RRID:BDSC_606006 | *w; gRNA-41F9-Qtg2.1-uDEH*$^{VK00018}$; in Materials and methods |
| Genetic reagent (*D. melanogaster*) | gRNA-41F11-Qtg2.1-uDEH | This study | RRID:BDSC_606007 | *w; gRNA-41F11-Qtg2.1-uDEH*$^{VK00018}$; in Materials and methods |
| Genetic reagent (*D. melanogaster*) | gRNA-42A4-Qtg2.1-tDEH | This study | | *w; gRNA-42A4-Qtg2.1-tDEH*$^{VK00018}$; in Materials and methods |
| Genetic reagent (*D. melanogaster*) | gRNA-42A4-Qtg2.1-tDES | This study | RRID:BDSC_606008 | *w; gRNA-42A4-Qtg2.1-tDES*$^{VK00018}$; in Materials and methods |
| Genetic reagent (*D. melanogaster*) | gRNA-80F5-Qtg2.1-tDEH | This study | RRID:BDSC_606014 | *w; gRNA-80F5-Qtg2.1-tDEH*$^{attP2}$; in Materials and methods |
| Genetic reagent (*D. melanogaster*) | gRNA-80C2-Qtg2.1-tDEH | This study | RRID:BDSC_606013 | *w; gRNA-80C2-Qtg2.1-tDEH*$^{attP2}$; in Materials and methods |
| Genetic reagent (*D. melanogaster*) | gRNA-80C1-Qtg2.1-tDEH | This study | RRID:BDSC_606012 | *w; gRNA-80C1-Qtg2.1-tDEH*$^{attP2}$; in Materials and methods |
| Genetic reagent (*D. melanogaster*) | gRNA-81F-Qtg2.1-tDEH | This study | RRID:BDSC_606020 | *w; gRNA-81F-Qtg2.1-tDEH*$^{VK00027}$; in Materials and methods |
| Genetic reagent (*D. melanogaster*) | gRNA-82A4-Qtg2.1-tDEH | This study | RRID:BDSC_606018 | *w; gRNA-82A4-Qtg2.1-tDEH*$^{VK00027}$; in Materials and methods |
| Genetic reagent (*D. melanogaster*) | gRNA-82C3-Qtg2.1-tDEH | This study | RRID:BDSC_606019 | *w; gRNA-82C3-Qtg2.1-tDEH*$^{VK00027}$; in Materials and methods |
| Genetic reagent (*D. melanogaster*) | gRNA-X2-Qtg2.1-tDES | This study | RRID:BDSC_606024 | *w; gRNA-X2-Qtg2.1-tDES*$^{attP18}$; in Materials and methods |
| Genetic reagent (*D. melanogaster*) | gRNA-20F1-Qtg2.1-tDES | This study | RRID:BDSC_606025 | *w; gRNA-20F1-Qtg2.1-tDES*$^{attP18}$; in Materials and methods |
| Genetic reagent (*D. melanogaster*) | gRNA-20F2-Qtg2.1-tDES | This study | RRID:BDSC_606026 | *w; gRNA-20F2-Qtg2.1-tDES*$^{attP18}$; in Materials and methods |
| Genetic reagent (*D. melanogaster*) | gRNA-101F1a-Qtg2.1-tDES | This study | RRID:BDSC_606776 | *w; gRNA-101F1a-Qtg2.1-tDES*$^{attP[102D]}$; in Materials and methods |
| Genetic reagent (*D. melanogaster*) | gRNA-101F1b-Qtg2.1-tDES | This study | RRID:BDSC_606777 | *w; gRNA-101F1b-Qtg2.1-tDES*$^{attP[102D]}$; in Materials and methods |

*Appendix 1 Continued on next page*

*Appendix 1 Continued*

| Reagent type (species) or resource | Designation | Source or reference | Identifiers | Additional information |
|---|---|---|---|---|
| Genetic reagent (*D. melanogaster*) | gRNA-101F1c-Qtg2.1-tDES | This study | RRID:BDSC_606778 | *w; gRNA-101F1c-Qtg2.1-tDES$^{attP[102D]}$; in Materials and methods* |
| Genetic reagent (*D. melanogaster*) | gRNA-40D2-nlsBFP | **Allen et al., 2021** | | *w; gRNA-40D2-nlsBFP$^{VK00037}$* |
| Genetic reagent (*D. melanogaster*) | gRNA-40D4-nlsBFP | **Allen et al., 2021** | | *w; gRNA-40D4-nlsBFP$^{VK00037}$* |
| Genetic reagent (*D. melanogaster*) | gRNA-40E1-nlsBFP | **Allen et al., 2021** | RRID:BDSC_606003 | *w; gRNA-40E1-nlsBFP$^{VK00037}$* |
| Genetic reagent (*D. melanogaster*) | gRNA-40D2-tub-IFP-HA | This study | | *w; gRNA-40D2-tub-IFP-HA$^{VK00037}$; in Materials and methods* |
| Genetic reagent (*D. melanogaster*) | gRNA-42A4-tub-BFP-HA | This study | RRID:BDSC_606011 | *w; gRNA-42A4-tub-BFP-HA$^{VK00018}$; in Materials and methods* |
| Genetic reagent (*D. melanogaster*) | gRNA-41F9-tub-BFP-HA | This study | RRID:BDSC_606010 | *w; gRNA-41F9-tub-BFP-HA$^{VK00018}$; in Materials and methods* |
| Genetic reagent (*D. melanogaster*) | gRNA-41F11-tub-BFP-HA | This study | RRID:BDSC_606009 | *w; gRNA-41F11-tub-BFP-HA$^{VK00018}$; in Materials and methods* |
| Genetic reagent (*D. melanogaster*) | gRNA-80F5-tub-BFP-HA | This study | RRID:BDSC_606017 | *w; gRNA-80F5-tub-BFP-HA$^{attP2}$; in Materials and methods* |
| Genetic reagent (*D. melanogaster*) | gRNA-80C2-tub-BFP-HA | This study | RRID:BDSC_606016 | *w; gRNA-80C2-tub-BFP-HA$^{attP2}$; in Materials and methods* |
| Genetic reagent (*D. melanogaster*) | gRNA-80C1-tub-BFP-HA | This study | RRID:BDSC_606015 | *w; gRNA-80C1-tub-BFP-HA$^{attP2}$; in Materials and methods* |
| Genetic reagent (*D. melanogaster*) | gRNA-81F-tub-BFP-HA | This study | RRID:BDSC_606021 | *w; gRNA-81F-tub-BFP-HA$^{VK00027}$; in Materials and methods* |
| Genetic reagent (*D. melanogaster*) | gRNA-82A4-tub-BFP-HA | This study | RRID:BDSC_606022 | *w; gRNA-82A4-tub-BFP-HA$^{VK00027}$; in Materials and methods* |
| Genetic reagent (*D. melanogaster*) | gRNA-82C3-tub-BFP-HA | This study | RRID:BDSC_606023 | *w; gRNA-82C3-tub-BFP-HA$^{VK00027}$; in Materials and methods* |
| Genetic reagent (*D. melanogaster*) | gRNA-X2-tub-BFP-HA | This study | RRID:BDSC_606028 | *w; gRNA-X2-tub-BFP-HA$^{attP18}$; in Materials and methods* |
| Genetic reagent (*D. melanogaster*) | gRNA-20F1-tub-BFP-HA | This study | RRID:BDSC_606779 | *w; gRNA-20F1-tub-BFP-HA$^{attP18}$; in Materials and methods* |
| Genetic reagent (*D. melanogaster*) | gRNA-20F2-tub-BFP-HA | This study | RRID:BDSC_606029 | *w; gRNA-20F2-tub-BFP-HA$^{attP18}$; in Materials and methods* |
| Genetic reagent (*D. melanogaster*) | gRNA-101F1a-tub-BFP-HA | This study | RRID:BDSC_606780 | *w; gRNA-101F1a-tub-BFP-HA$^{attP[102D]}$; in Materials and methods* |
| Genetic reagent (*D. melanogaster*) | gRNA-101F1b-tub-BFP-HA | This study | RRID:BDSC_606781 | *w; gRNA-101F1b-tub-BFP-HA$^{attP[102D]}$; in Materials and methods* |
| Genetic reagent (*D. melanogaster*) | gRNA-101F1c-tub-BFP-HA | This study | RRID:BDSC_606782 | *w; gRNA-101F1c-tub-BFP-HA$^{attP[102D]}$; in Materials and methods* |
| Genetic reagent (*D. melanogaster*) | pxn-Gal4 | **Han et al., 2014** | | |
| Genetic reagent (*D. melanogaster*) | UAS-CD4-tdTom | **Han et al., 2011** | RRID:BDSC_35841 | *UAS-CD4-tdTom$^{7M1}$* |

*Appendix 1 Continued on next page*

*Appendix 1 Continued*

| Reagent type (species) or resource | Designation | Source or reference | Identifiers | Additional information |
|---|---|---|---|---|
| Genetic reagent (*D. melanogaster*) | hh-Cas9 | *Poe et al., 2019* | | R28E04-Cas9$^{GA}$ |
| Genetic reagent (*D. melanogaster*) | zk-cas9 | *Allen et al., 2021* | | zk-Cas9$^{VK00037}$ |
| Genetic reagent (*D. melanogaster*) | hs-cas9 | *Garcia-Marques et al., 2020* | | |
| Genetic reagent (*D. melanogaster*) | Gal4$^{21-7}$ | *Han et al., 2011* | | |
| Genetic reagent (*D. melanogaster*) | UAS-MApHS | *Han et al., 2014* | | UAS-MApHS$^{VK00019}$ |
| Genetic reagent (*D. melanogaster*) | RabX4-Gal4 | Bloomington *Drosophila* Stock Center | RRID:BDSC_51602 | |
| Genetic reagent (*D. melanogaster*) | R38F11-Gal4 | Bloomington *Drosophila* Stock Center | RRID:BDSC_50014 | R38F11-Gal4attP2 |
| Genetic reagent (*D. melanogaster*) | ey-Cas9 | *Ji et al., 2022* | | ey-Cas9$^{VK00005}$ |
| Genetic reagent (*D. melanogaster*) | vas-cas9 | *López Del Amo et al., 2022* | | |
| Genetic reagent (*D. melanogaster*) | tubP(FRT.stop)Gal4 UAS-Flp UAS-mCD8::GFP | *Koreman et al., 2021* | | |
| Genetic reagent (*D. melanogaster*) | α-Catenin-GFP | Bloomington *Drosophila* Stock Center | RRID:BDSC_58787 | w[*]; P{w[+mC]=UAS-alpha-Cat.T:GFP.sg}3/CyO |
| Genetic reagent (*D. melanogaster*) | gcm-Cas9 | *Chen et al., 2024* | | |
| Genetic reagent (*D. melanogaster*) | repo-Gal4 | Bloomington *Drosophila* Stock Center | RRID:BDSC_7415 | |
| Genetic reagent (*D. melanogaster*) | Act5C-Cas9 | Bloomington *Drosophila* Stock Center | RRID:BDSC_54590 | Act5C-Cas9.P |
| Genetic reagent (*D. melanogaster*) | vGlut$^{SS1}$ | Bloomington *Drosophila* Stock Center | RRID:BDSC_91246 | |
| Genetic reagent (*D. melanogaster*) | brp$^{d09839}$ | Bloomington *Drosophila* Stock Center | RRID:BDSC_85508 | |
| Genetic reagent (*D. melanogaster*) | EcR$^{M554fs}$ | Bloomington *Drosophila* Stock Center | RRID:BDSC_4894 | |
| Genetic reagent (*D. simulans*) | Lhr$^1$ (*D. simulans*) | *Brideau et al., 2006* | | |
| Genetic reagent (*D. melanogaster*) | Df(4)ED6380 | Bloomington *Drosophila* Stock Center | RRID:BDSC_602664 | |
| Recombinant DNA reagent | pAC-U63-QtgRNA2.1-ubiGal80(DE)-His2AV (plasmid) | This study | | in Materials and methods |
| Recombinant DNA reagent | pAC-U63-QtgRNA2.1-tubGal80(DE)-His2AV (plasmid) | This study | | in Materials and methods |
| Recombinant DNA reagent | pAC-U63-QtgRNA2.1-tubGal80(DE)-SV40 (plasmid) | This study | | in Materials and methods |
| Recombinant DNA reagent | pAC-U63-QtgRNA2.1-tubBFP(HA) (plasmid) | This study | | in Materials and methods |
| Recombinant DNA reagent | pAC-U63-QtgRNA2.1-tub-miRFP680-T2A-HO1(HA) (plasmid) | This study | | in Materials and methods |
| Recombinant DNA reagent | pAC-U63-QtgRNA2.1–8 R (plasmid) | *Koreman et al., 2021* | RRID:Addgene 170514 | |
| Recombinant DNA reagent | pENTR221-tubP (plasmid) | *Chen et al., 2025* | | |
| Recombinant DNA reagent | pAPIC-PHCS (plasmid) | *Han et al., 2011* | | |

*Appendix 1 Continued on next page*

*Appendix 1 Continued*

| Reagent type (species) or resource | Designation | Source or reference | Identifiers | Additional information |
|---|---|---|---|---|
| Recombinant DNA reagent | pAC-U63-QtgRNA2.1-BR (plasmid) | *Koreman et al., 2021* | RRID:Addgene 170513 | |
| Software | Fiji | https://fiji.sc/ | RRID:SCR_002285 | |
| Software | R | https://www.r-project.org/ | RRID:SCR_001905 | |
| Software | Adobe Photoshop | Adobe | RRID:SCR_014199 | |
| Software | Adobe Illustrator | Adobe | RRID:SCR_010279 | |
| Chemical compound | DAPI (4',6-Diamidino-2-Phenylindole) | Life Technology | RRID:62248 | 1:1000 dilution |
| Antibody | anti-Elav 7E8A10 (Rat monoclonal) | Developmental Studies Hybridoma Bank | RRID:AB_528218 | 1:50 dilution |
| Antibody | anti-Repo 8D12 (Mouse monoclonal) | Developmental Studies Hybridoma Bank | RRID:AB_528448 | 1:50 dilution |
| Antibody | anti-Brp nc82 (Mouse monoclonal) | Developmental Studies Hybridoma Bank | RRID:AB_2314866 | 1:100 dilution |
| Antibody | anti-vGluT (Rabbit polyclonal) | *Chen et al., 2024* | | 1:200 dilution |
| Antibody | anti-HA 12CA5 (mouse monoclonal) | Sigma Aldrich | Roche 11583816001 | 1:100 dilution |
| Antibody | anti-HRP conjugated with Cy3 (goat polyclonal) | Jackson ImmunoResearch | RRID:AB_2338952 | 1:200 dilution |
| Antibody | anti-mouse secondary antibody conjugated with Cy5 (donkey polyclonal) | Jackson ImmunoResearch | RRID:AB_2340820 | 1:400 dilution |
| Antibody | anti-rabbit secondary antibody conjugated with Cy5 (donkey polyclonal) | Jackson ImmunoResearch | RRID:AB_2340607 | 1:400 dilution |
| Antibody | anti-rat secondary antibody conjugated with Cy5 (donkey polyclonal) | Jackson ImmunoResearch | RRID:AB_2340672 | 1:400 dilution |
| Commercial assay or kit | T4 ligase | New England Biolabs Inc. | #M0202 | |
| Commercial assay or kit | NEBuilder HiFi DNA Assembly Master Mix | New England Biolabs Inc. | #E2621 | |

