## [Editor Report · eLife Assessment]

The study showcases a significant and **important** enhancement of the MAGIC transgenesis method, by extending it genome-wide to all chromosomes. The authors provide **compelling** evidence to demonstrate that the MAGIC mosaic clones can be generated for genes from all, including the 4th chromosome. With this toolkit extension, the method is set to complement the classical FRT/Flp recombination system for gene manipulation in flies.

---

## [Referee Report · Reviewer #1 (Public review)]

Summary:

In this manuscript, Shen et al. have improved upon the mitotic clone analysis tool MAGIC that their lab previously developed. MAGIC uses CRISPR/Cas9-mediated double-stranded breaks to induce mitotic recombination. The authors have replaced the sgRNA scaffold with a more effective scaffold to increase clone frequency. They also introduced modifications to positive and negative clonal markers to improve signal-to-noise and mark the cytoplasm of the cells instead of the nuclei. The changes result in increase in clonal frequencies and marker brightness. The authors also generated the MAGIC transgenics to target all chromosome arms and tested the clone induction efficacy.

Strengths:

MAGIC is a mitotic clone generation tool that works without prior recombination to special chromosomes (e.g., FRT). It can also generate mutant clones for genes for which the existing FRT lines could not be used (e.g., the genes that are between the FRT transgene and the centromere).

This manuscript does a thorough job in describing the method and provides compelling data that support improvement over the existing method.

---

## [Referee Report · Reviewer #2 (Public review)]

Summary:

In this study, the authors present the latest improvement of their previously published methods, pMAGIC and nMAGIC, which can be used to engineer mosaic gene expression in wild-type animals and in a tissue-specific manner. They address the main limitation of MAGIC, the lack of gRNA-marker transgenes, which has hampered the broader adoption of MAGIC in the fly community. To do so, they create an entire toolkit of gRNA markers for every Drosophila chromosome and test them across a range of different tissues and in the context of making Drosophila species hybrid mosaic animals. The study provides a significant and broadly useful improvement compared to earlier versions, as it broadens the use-cases for transgenic manipulation with MAGIC to virtually any subfield of Drosophila cell biology.

Strengths:

Major improvements to MAGIC were made in terms of clone induction efficiency and usability across the Drosophila model system, including wild-type genotypes and the use in non-melanogaster species.

Notably, mosaic mutants can now be created for genes residing on the 4th chromosome, which is exciting and possibly long-awaited by 4th chromosome gene enthusiasts.

Selection of the standard set of gRNA markers was done thoughtfully, using non-repetitive conserved and unique sequences.

The authors demonstrate that MAGIC can be used easily in the context of interspecific hybrids. I believe this is a great advancement for the Drosophila community, especially for evolutionary biologists, because this may allow for easy access to mechanistic, tissue-specific insight into the process of a range of hybrid incompatibilities, an important speciation process that is normally difficult to study at the level of molecular and cell biology.

In the same way, because it is not limited to usage in any particular genetic background, genome-wide MAGIC can be potentially used in wild-type genotypes relatively easily. This is exciting, especially because natural genetic diversity is rarely investigated more mechanistically and at the scale/resolution of cells or specific tissues. Now, one can ask how a particular naturally occurring allele influences cell physiology compared to another (control) while keeping the global physiological context of the particular genetic background largely intact.

---

## [Referee Report · Reviewer #3 (Public review)]

Summary:

In the manuscript by Shen, Yeung, and colleagues, the authors generate an improved and expanded Mosaic analysis by gRNA-induced crossing-over (MAGIC) toolkit for use in making mosaic clones in Drosophila. This is a clever method by which mitotic clones can be induced in dividing cells by using CRISPR/Cas9 to generate double-strand breaks at specific locations that induce crossing over at those locations. This is conceptually similar to previous mosaic methods in flies that utilized FRT sites that had been inserted near centromeres along with heat-shock inducible FLPase. The advantage of the MAGIC system is that it can be used along with chromosomes lacking FRT sites already introduced, such as those found in many deficiency collections or in EMS mutant lines. It may also be simpler to implement than FRT-based mosaic systems. There are two flavors of the MAGIC system: nMAGIC and pMAGIC. In nMAGIC, the main constituents are a transgene insertion that contains gRNAs that target DNA near the centromere, along with a fluorescent marker. In pMAGIC, the main constituents are a transgenic insertion that contains gRNAs that target DNA near the centromere, along with ubiquitous expression of GAL80. As such, nMAGIC can be used to generate clones that are not labelled, whereas pMAGIC (along with a GAL4 line and UAS-marker) can be used much like MARCM to positively label a clone of cells. This manuscript introduces MAGIC transgenic reagents that allow all 4 chromosomes to be targeted. They demonstrate its use in a variety of tissues, including with mutants not compatible with current FLP/FRT methods, and also show it works well in tissues that prove challenging for FLP/FRT mosaic analyses (such as motor neurons). They further demonstrate that it can be used to generate mosaic clones in non-melanogaster hybrid tissues. Overall, this work represents a valuable improvement to the MAGIC method that should promote even more widespread adoption of this powerful genetic technique.

Strengths:

(1) Improves the design of the gRNA-marker by updating the gRNA backbone and also the markers used. GAL80 now includes a DE region that reduces the perdurance of the protein and thus better labeling of pMAGIC clones. The data presented to demonstrate these improvements is rigorous and of high quality.

(2) Introduces a toolkit that now covers all chromosome arms in Drosophila. In addition, the efficiency of 3 target different sites is characterized for each chromosome arm (e.g., 3 different gRNA-Marker combinations), which demonstrate differences in efficiency. This could be useful to titrate how many clones an experimenter might want (e.g., lower efficiency combinations might prove advantageous).

(3) The manuscript is well written and easy to follow. The authors achieved their aims of creating and demonstrating MAGIC reagents suitable for mosaic analysis of any Drosophila chromosome arm.

(4) The MAGIC method is a valuable addition to the Drosophila genetics toolkit, and the new reagents described in this manuscript should allow it to become more widely adopted.

Comments on revised version:

The authors have done a great job addressing reviewer concerns with the addition of updated figures, new experiments, and changes to the manuscript. I am supportive of this version and agree with the updated assessment.

---

## [Author Response]

The following is the authors’ response to the original reviews.

We greatly appreciate the reviewers’ constructive comments and have followed their recommendations to improve our manuscript. These improvements include additional experiments, new analyses, and a rewriting of the text. We believe these changes significantly improved the paper and hope the editor and the reviewers agree. The following is a summary of the major changes made and our point-by-point response to reviewers’ comments.

Summary of major changes:

(1) Expanded labeling options: We generated a new nMAGIC vector containing miRFP680 as an infrared fluorescent protein (IFP) marker. We used gRNA-40D2(IFP) to demonstrate clones labeled by this marker in the wing imaginal disc (Figure 1M). This vector is available via Addgene for the generation of new gRNA-markers with our recommended or customer-designed gRNA target sequences.

(2) Validated Gal80 potency: We provide new data in Figure 1E demonstrating complete suppression of *pxn-Gal4>CD4-tdTom* by *tub-GAL80-DE-SV40*. The exact transgenes used in the comparisons are clarified in the figure and figure legend.

(3) Verified clone fitness: We compared the sizes of nMAGIC twin spots in wing discs and found no intrinsic growth or viability bias between marker/marker and WT/WT clones (Figure 1O).

(4) Methodological Schematics: We added supplemental figures to Figure 1 to illustrate the principle of MAGIC, the difference between pMAGIC and nMAGIC, and an example of pMAGIC crossing scheme.

(5) Inducible induction: We provide new data (Figure 3J-K’) showing the induction of sparse neuronal clones in the adult brain by heat shock (hs)-Cas9.

(6) We revised texts to incorporate all other recommendations suggested by the reviewers. We also made other small changes to the manuscript to improve its readability.

**Public Reviews:**

**Reviewer #1 (Public review):**
Summary:In this manuscript, Shen et al. have improved upon the mitotic clone analysis tool MAGIC that their lab previously developed. MAGIC uses CRISPR/Cas9-mediated double-stranded breaks to induce mitotic recombination. The authors have replaced the sgRNA scaffold with a more effective scaffold to increase clone frequency. They also introduced modifications to positive and negative clonal markers to improve signal-to-noise and mark the cytoplasm of the cells instead of the nuclei. The changes result in increase in clonal frequencies and marker brightness. The authors also generated the MAGIC transgenics to target all chromosome arms and tested the clone induction efficacy.Strengths:MAGIC is a mitotic clone generation tool that works without prior recombination to special chromosomes (e.g., FRT). It can also generate mutant clones for genes for which the existing FRT lines could not be used (e.g., the genes that are between the FRT transgene and the centromere).This manuscript does a thorough job in describing the method and provides compelling data that support improvement over the existing method.Weaknesses:It would be beneficial to have a greater variety of clonal markers for nMAGIC. Currently, the only marker is BFP, which may clash with other genetic tools (e.g., some FRET probes) depending on the application. It would be nice to have far-red clonal markers.

We thank the reviewer for the positive comments about our study. We agree with the reviewer that adding a far-red option for nMAGIC increases the flexibility of this method. We replaced the BFP coding sequence in the nMAGIC cloning vector pAC-U63-QtgRNA2.1-tubBFP(HA) with that of miRFP680-T2A-HO1. We then used the resulting cloning vector to make a gRNA-40D2(IFP) transgene and tested it in the wing disc. Result showing clones in the wing disc are now in Figure 1M. The new cloning vector, along with others reported in our study, are available from Addgene.

**Reviewer #2 (Public review):**
Summary:In this study, the authors present the latest improvement of their previously published methods, pMAGIC and nMAGIC, which can be used to engineer mosaic gene expression in wild-type animals and in a tissue-specific manner. They address the main limitation of MAGIC, the lack of gRNA-marker transgenes, which has hampered the broader adoption of MAGIC in the fly community. To do so, they create an entire toolkit of gRNA markers for every Drosophila chromosome and test them across a range of different tissues and in the context of making Drosophila species hybrid mosaic animals. The study provides a significant and broadly useful improvement compared to earlier versions, as it broadens the use-cases for transgenic manipulation with MAGIC to virtually any subfield of Drosophila cell biology.Strengths:Major improvements to MAGIC were made in terms of clone induction efficiency and usability across the Drosophila model system, including wild-type genotypes and the use in non-melanogaster species.Notably, mosaic mutants can now be created for genes residing on the 4th chromosome, which is exciting and possibly long-awaited by 4th chromosome gene enthusiasts.Selection of the standard set of gRNA markers was done thoughtfully, using non-repetitive conserved and unique sequences.The authors demonstrate that MAGIC can be used easily in the context of interspecific hybrids. I believe this is a great advancement for the Drosophila community, especially for evolutionary biologists, because this may allow for easy access to mechanistic, tissue-specific insight into the process of a range of hybrid incompatibilities, an important speciation process that is normally difficult to study at the level of molecular and cell biology.In the same way, because it is not limited to usage in any particular genetic background, genome-wide MAGIC can be potentially used in wild-type genotypes relatively easily. This is exciting, especially because natural genetic diversity is rarely investigated more mechanistically and at the scale/resolution of cells or specific tissues. Now, one can ask how a particular naturally occurring allele influences cell physiology compared to another (control) while keeping the global physiological context of the particular genetic background largely intact.Weaknesses:It is not entirely clear how functionally non-critical regions were evaluated, besides that they are selected based on conservation of sequence between species. It may be useful to directly test the difference in viability or other functionally relevant phenotype for flies carrying different markers. Similarly, the frequency of off-targets could be investigated or documented in a bit more detail, especially if one of the major use-cases is meant for naturally derived, diverse genetic backgrounds. It is, at the moment, unclear how consistently the clones are induced for each new gRNA marker across different WT genetic backgrounds, for example, a set of DGRP genotypes, which could be highly useful information for future users.

We thank the reviewer for the positive comments about our study. The reviewer raises an excellent point regarding the consistency of clone induction and potential background effects in diverse genetic backgrounds. As a standard step in building the MAGIC kit, we tested all gRNA-marker transgenes with the Cas9-LEThAL assay (Poe et al., Genetics, 2019), in which the gRNA-marker transgene was crossed to *lig4 Act5C-Cas9* homozygotes. All crosses led to viable and apparently healthy female progeny, suggesting that ubiquitously mutating the chosen gRNA targeting sites does not cause obvious defects.

For standard mutant analysis, we recommend researchers to use a well-characterized wildtype chromosome as a negative control. For studies utilizing diverse wildtype backgrounds where a standard control chromosome is inapplicable (e.g., DGRP screens), we recommend an internal validation strategy: researchers should confirm their key phenotypic findings by inducing clones with a second, independent gRNA-marker located on the same chromosomal arm (e.g., comparing clones induced by gRNA-40D2 vs. gRNA-40D4). This ensures that any observed phenotypes or variations in clone induction are linked to the selected genetic background rather than an off-target artifact or target-site specific effect.

We admit that the above approach may not resolve concerns about off-targets. Performing deep sequencing to map empirical off-targets for all 34 gRNA pairs across multiple genetic backgrounds is experimentally prohibitive for a toolkit resource. However, our *in silico* selection pipeline strictly required target sequences to be unique within the *D. melanogaster* genome to mathematically minimize off-target probability. In addition, our requirement that target sequences be conserved in closely related *Drosophila* species acts as a stringent filter against intraspecies variation. Sequences conserved across species are subject to purifying selection, substantially reducing the likelihood that SNPs within the DGRP lines will disrupt the PAM or seed sequences required for Cas9 induction.

**Reviewer #3 (Public review):**
Summary:In the manuscript by Shen, Yeung, and colleagues, the authors generate an improved and expanded Mosaic analysis by gRNA-induced crossing-over (MAGIC) toolkit for use in making mosaic clones in Drosophila. This is a clever method by which mitotic clones can be induced in dividing cells by using CRISPR/Cas9 to generate double-strand breaks at specific locations that induce crossing over at those locations. This is conceptually similar to previous mosaic methods in flies that utilized FRT sites that had been inserted near centromeres along with heat-shock inducible FLPase. The advantage of the MAGIC system is that it can be used along with chromosomes lacking FRT sites already introduced, such as those found in many deficiency collections or in EMS mutant lines. It may also be simpler to implement than FRT-based mosaic systems. There are two flavors of the MAGIC system: nMAGIC and pMAGIC. In nMAGIC, the main constituents are a transgene insertion that contains gRNAs that target DNA near the centromere, along with a fluorescent marker. In pMAGIC, the main constituents are a transgenic insertion that contains gRNAs that target DNA near the centromere, along with ubiquitous expression of GAL80. As such, nMAGIC can be used to generate clones that are not labelled, whereas pMAGIC (along with a GAL4 line and UAS-marker) can be used much like MARCM to positively label a clone of cells. This manuscript introduces MAGIC transgenic reagents that allow all 4 chromosomes to be targeted. They demonstrate its use in a variety of tissues, including with mutants not compatible with current FLP/FRT methods, and also show it works well in tissues that prove challenging for FLP/FRT mosaic analyses (such as motor neurons). They further demonstrate that it can be used to generate mosaic clones in non-melanogaster hybrid tissues. Overall, this work represents a valuable improvement to the MAGIC method that should promote even more widespread adoption of this powerful genetic technique.Strengths:(1) Improves the design of the gRNA-marker by updating the gRNA backbone and also the markers used. GAL80 now includes a DE region that reduces the perdurance of the protein and thus better labeling of pMAGIC clones. The data presented to demonstrate these improvements is rigorous and of high quality.(2) Introduces a toolkit that now covers all chromosome arms in Drosophila. In addition, the efficiency of 3 target different sites is characterized for each chromosome arm (e.g., 3 different gRNA-Marker combinations), which demonstrate differences in efficiency. This could be useful to titrate how many clones an experimenter might want (e.g., lower efficiency combinations might prove advantageous).(3) The manuscript is well written and easy to follow. The authors achieved their aims of creating and demonstrating MAGIC reagents suitable for mosaic analysis of any Drosophila chromosome arm.(4) The MAGIC method is a valuable addition to the Drosophila genetics toolkit, and the new reagents described in this manuscript should allow it to become more widely adopted.Weaknesses:(1) The MAGIC method might not be well known to most readers, and the manuscript could have benefited from schematics introducing the technique.

We thank the reviewer for the positive evaluation of our study and for making this kind suggestion. We have added diagrams that explain the principle of MAGIC and the difference between pMAGIC and nMAGIC in Figure 1 - Figure Supplement 1.

(2) Traditional mosaic analyses using the FLP/FRT system have strongly utilized heat-shock FLPase for inducible temporal control over mitotic clones, as well as a way to titrate how many clones are induced (e.g., shorter heat shocks will induce fewer clones). This has proven highly valuable, especially for developmental studies. A heat-shock Cas9 is available, and it would have been beneficial to determine the efficiency of inducing MAGIC clones using this Cas9 source.

We thank the reviewer for suggesting this experiment. We agree that demonstrating inducible clone induction in the adult brain is an effective way for people to compare MAGIC with the MARCM method they are probably more familiar with. We used a heat shock Cas9 developed by the Tzumin Lee group (Chen et al., Development, 2020) to experiment with clone induction, and the results are shown in the new Figure 3 (K and J). We show that, with a pan-neuronal Gal4, heat shock during the wandering 3rd instar larval stage induced more clones than during the pupal stage, and the later heat shock readily produced sparsely labeled neurons whose single-cell morphology can be easily visualized.

**Recommendations for the authors:**

**Reviewing Editor Comments:**
The following are some consolidated review remarks after discussions amongst all three reviewers:The reviewers feel the evidence level could be raised from 'convincing' to 'compelling' if the following key (and partially shared) suggestions by the reviewers are followed adequately:(1) Expand labeling options for nMAGIC, which is currently just a BFP marker. This would increase the utility of the method. A far-red marker would be very helpful. Could the authors just do this for one chromosome arm and make the reagent available for others to generate other chromosome arms?

We agree with the editor and reviewers that adding a far-red option for nMAGIC increases the flexibility of this method. We replaced the BFP coding sequence in the nMAGIC cloning vector pAC-U63-QtgRNA2.1-tubBFP(HA) with that of miRFP680-T2A-HO1. We then used the resulting cloning vector to make a gRNA-40D2(IFP) transgene and tested it in the wing disc. Result showing clones in the wing disc are now in Figure 1M. The new cloning vector, along with others reported in our study, will be available from Addgene.

(2) Verify that destabilized GAL80 is potent enough to suppress GAL4. Repeat Figure 1C-E with tub-GAL80-DE-SV40.

We replaced the experiment using gRNA-42A4-tDES, which successfully achieved complete suppression of pxn>CD4-tdTom (Figure 1E).

(3) Concern about the health of the induced mitotic clones. This is an important consideration, but the reviewers were not sure what the necessary experiments would be. To gauge twin-spot clone sizes? Please address.

We agree that clone fitness is an important consideration for MAGIC experiments. To test it, we generated WT clones in the wing imaginal disc using nMAGIC and quantified the sizes of the twin spots (*BFP/BFP* and *WT/WT* clones). Our results show that there is no statistical difference between these two types of clones. Thus, there is no intrinsic growth disadvantage to either type of mitotic clones generated by MAGIC.

(4) Include a schematic of the MAGIC method as Figure 1 or add it to Figure 1. Many may not be familiar with the method, so to promote its adoption, the authors should clearly introduce the MAGIC method in this paper (and not rely on readers to go to previous publications). For this paper to become a MAGIC reference paper, it should be self-contained.

We thank the reviewers for this suggestion. We have added diagrams that explain the principle of MAGIC and the difference between pMAGIC and nMAGIC in Figure 1 - Figure Supplement 1.

(5) Determine the utility of using a hs-Cas9 line for temporal induction of MAGIC clones. This is a traditional method for mitotic clone induction (with hsFLP/FRTs), and its use with the MAGIC system (especially pMAGIC) could also make it more attractive, especially to label small populations of neurons born at known times. To this point, the authors could generate pMAGIC clones using hs-Cas9 for commonly used adult target neurons, such as projection neurons, central complex neurons, or mushroom body neurons. The method to label small numbers of these adult neurons is well worked out with known GAL4 lines, and demonstrating that pMAGIC could have similar results would capture the attention of many not familiar with the pMAGIC method.

We agree that demonstrating inducible clone induction in the adult brain is an effective way for people to compare MAGIC with the MARCM method they are probably more familiar with. We used a heat shock Cas9 developed by the Tzumin Lee group (GarciaMarques, Espinosa-Medina et al. 2020) to experiment with clone induction, and the results are shown in the new Figure 3 (J-K’). We show that, with a pan-neuronal Gal4, heat shock during wandering 3rd instar larval stage induced more clones than during the pupal stage, and the later heat shock readily produced sparsely labeled neurons whose single-cell morphology can be easily visualized.

**Reviewer #1 (Recommendations for the authors)**:This is a marked improvement over the existing methods that the authors' lab has previously generated. It will be a nice addition to the Drosophila genetic tool kit after minor revisions.

We appreciate the reviewer’s recognition of the new tools we developed.

Minor issues:(1) In the data in Figures 1G and H, it is not ideal to compare the effect of different modifications on two different transgenes. uH and uDEH are compared in gRNA-40D2, whereas uDEH, tDEH, and tDES are compared in gRNA-42A4. If the transgenics are already available, it would be better to compare the uH, uDEH, tDEH, and tDES on either gRNA-40D2 or gRNA-42A4.

We appreciate the reviewer’s concern. These transgenes were developed during different phases of this project. We first adopted the uDEH design during improvement of gRNA40D2, which solved both the leaky activity of pxn-Gal4 and dim epidermal clones. However, when we tried to expand this design to 2R (such as 42A4), we found that the clones were still too dim (probably due to positional effects). Thus, we next used uDEH in gRNA-42A4 as a base for further improvements. We did not make a uH version for gRNA-42A4 because we already knew that it is inferior to uDEH. Because of this history, we did not have the full set for gRNA42A4.

Despite the lack of uH for gRNA-42A4, we believe our comparisons of different designs are still valid, given that uH and uDEH were compared with identical sequences elsewhere in the transgenic vector (including the gRNA target sequence) and in the identical insertion site.

(2) It is not clear whether the authors tested destabilized Gal80 is potent to suppress Gal4 (e.g., in suppressing pxn>CD4-tdTom in hemocytes). The results in Figure 1C-E should be repeated with tub-Gal80-DE-SV40.

We apologize for omitting the transgene identities in these experiments. We have redone the experiment using gRNA-42A4-tDES and updated the figures to clearly indicate which transgenes were used.

(3) The difference in sgRNA scaffolds can be better explained in the text. The explanation here is very bare bones and reads like jargon. (i.e., changing F+E gRNA scaffold with gRNA2.1 scaffold is not a sufficient explanation).

We have added more explanations to the differences between the scaffolds as suggested.

(4) The stocks should be sent to Bloomington Stock Center to ensure widespread adoption of the method. This includes the Cas9 lines that are generated and used.

It is our plan to freely share the reagents developed in this study with the community. Most of the fly lines are already available at Bloomington (here and here). We are in the process of depositing the remaining ones to BDSC.

In conclusion, this is a nicely written manuscript that improves currently available tools and should be of interest to the readership of this journal.
**Reviewer #2 (Recommendations for the authors):**
Typos spotted:Line 163 issues -> tissuesLine 613 significance -> significant

We thank the reviewer for catching these typos. We have corrected them.

**Reviewer #3 (Recommendations for the authors):**
This is a welcome update to the MAGIC system, which is a brilliant method that has not been as widely adopted as it should be. The authors validate and introduce updates to this system to increase clonal efficiency and more robust labeling (for both pMAGIC and nMAGIC). The data presented are robust and convincing.

We appreciate the reviewer’s positive comments about our study.

Suggestions to improve the presentation and adoption of this work:(1) The MAGIC system might not be well known, and the manuscript would have benefited from an introductory schematic of how the system works. I realize this was already done in the PLoS Biology paper, but the authors should not assume readers will know that paper, or be willing to look it up. So a standalone schematic, as Figure 1, or something added to Figure 1, would greatly aid in understanding how this system works and what the new updates are doing.

We thank the reviewer for this kind suggestion. We have added diagrams that explain the principle of MAGIC and the difference between pMAGIC and nMAGIC in Figure 1 - figure supplement 1.

(2) There were many instances where abbreviations were not clearly defined, especially in the Figures and Figure legends. The main text is well-written, and while the information is in there, it is beneficial when the Figures and Figure legends can stand alone. For example:(a) Figure 1. DE, not defined in the Figure or Figure legend.(b) Figure 1. 'p' and 'n' not defined in the Figure legend.(c) The different Cas9 lines or GAL4 lines used-a brief description of their expression patterns might be helpful in the legend. E.g., zk-Cas9, vas-Cas9, gcm-Cas9, R38F11-GAL4, RabX4Gal4.

We apologize for omitting the details mentioned. They have been added to the figures and figure legends.

(3) "Traditional" mosaic analyses took advantage of hsFLP for inducible induction and to control the number of mitotic clones that were induced. A hs-Cas9 line does exist (as correctly pointed out by the authors), and it would be a valuable addition if the authors tested the utility of this reagent with the MAGIC system. Many possible adopters may not like the idea that an alwayson Cas9 line is used, which could result in too many clones, especially if one wanted to label very few cells. Granted, one could use a 'worse' gRNA-Marker line as mentioned in the manuscript, but this might still be hard to titrate, as well as an inducible system that uses a heatshock promoter. A hs promoter is especially useful for birthdating cells during development.

We thank the reviewer for suggesting this experiment. We agree that demonstrating inducible clone induction in the adult brain is an effective way for people to compare MAGIC with the MARCM method they are probably more familiar with. We used a heat shock Cas9 developed by the Tzumin Lee group (Chen et al., Development, 2020) to experiment with clone induction, and the results are shown in the new Figure 3 (K and J). We show that, with a panneuronal Gal4, heat shock during wandering 3rd instar larval stage induced more clones than during the pupal stage, and the later heat shock readily produced sparsely labeled neurons whose single-cell morphology can be easily visualized.

(4) Lines 61-63. "However, most of these mutant chromosomes cannot be analyzed by traditional mosaic techniques due to the lack of FRT sites or incompatibility with the FRT/Flp system." It might also be worth mentioning that recombining existing reagents (e.g., mutants, etc) onto an FRT chromosome can be labor and time-intensive. A brilliant advantage of MAGIC is that it can be used with any existing stock, such as from classical EMS mutant screens, Df screens (as pointed out), etc. So the more the authors can emphasize a new way of thinking (e.g, you don't need to recombine your mutant of interest onto an FRT stock before you can get started), the better!

We thank the reviewer for this kind suggestion. As suggested, we have expanded our introduction and discussion to emphasize the advantages of the MAGIC system over traditional mosaic techniques.

(5) One incredible advantage of the MAGIC system is that it can direct where recombination occurs. So if one had two mutations on a chromosome arm, it could be possible to make the most distal homozygous mutant while the other remains heterozygous. This is not possible with current FRT-based methods. It's not necessary to demonstrate this, but perhaps the authors could mention it as a possible next step? This was somewhat implied by lines 66-67 "In comparison, MAGIC can potentially be used to study these genes because the crossover site in MAGIC can be flexibly defined by users".

Again, we thank the reviewer for this nice suggestion. We have added this point to the discussion.

(6) How stable are the MAGIC lines? If gRNA (with Cas9 expressed) induced a germline mutation of the target site, the MAGIC line would break down. How often is this observed? Some mention of this would be appreciated, especially to end users, if caution is necessary and gRNA-marker stocks should not be maintained in the same flies as an x-Cas9 line.

The reviewer made a very important point. Keeping gRNA and Cas9 in the same strain will risk mutating the target sequence in the germline, if the Cas9 has any activity in the germline. Thus, it is not recommended to keep gRNA and Cas9 in the same flies over multiple generations. For MAGIC experiments, this concern is lessened because by crossing gRNA + Cas9 flies to another strain containing the chromosome of interest, clones can still be induced (possibly with less efficiency) because the chromosome of interest is still cuttable by Cas9. Nevertheless, to address this concern, we have recently developed anti-CRISPR tools to suppress Cas9 activity in such strains. These tools will be reported in a separate study.

In the revised manuscript, we added this point in Discussion to caution users.

(7) Line 157, "identify efficient gRNAs for every chromosomal arm.". What is considered "efficient"? Is this quantifiable? Eg., >= 10 clones.

Thanks for pointing this out! “Efficient” is an arbitrary evaluation, as different experiments may require different efficiencies. But operationally, we consider any gRNA that can generate >= 10 neuronal clones per larva as being efficient. We have clarified it in the text.

(8) Line 163, "highly packed _issues_ such as the brain"; spelling, should be "tissues"

Thanks for catching this typo. It has been corrected.

(9) The authors use ey-Cas9 for their demonstration of adult brain labeling. Additional adult brain examples would increase exposure of this method and attract wider attention- targeting structures that have been well characterized, such as projection neurons (GH146-GAL4), central complex, mushroom bodies, etc. Especially if hs-Cas9 could be utilized to mimic previous MARCM clones (for example).

We thank the reviewer for suggesting heat shock-induced clones in the adult brain. We have conducted the experiment as explained above and shown in Figure 3J-3K’. We showed a single neuronal clone that resembles lateral horn Leucokinin neurons.

(10) Line 216, "Despite these advances, existing mutations on FRT-lacking 4th chromosomes still cannot be analyzed by the FRT/Flp system." For context, it might be worth pointing out that meiotic recombination is exceedingly rare on the 4th chromosome, which means it is practically impossible to recombine existing 4th chromosome mutations onto an FRT chromosome.

We thank the reviewer for this kind suggestion. We have added a note about the difficulty of recombining FRT onto the 4th chromosome.

(11) Figure 2 legend. What is the full genotype for D and E? eg, what is RabX4>MApHS?

We apologize for being brief with the details. RabX4-Gal4 is a pan-neuronal driver. UAS-MApHS is a membrane fluorescent marker (UAS-pHluorin-CD4-tdTom). The genotypes have been added to the figure legend.

(12) It would be good to include the Bloomington Stock numbers for the MAGIC toolkit, especially in Table 1. And include an HTML reference to their MAGIC page at Bloomington(https://bdsc.indiana.edu/stocks/misc/magic.html).

Thank you for this suggestion! We have done as suggested.

(13) Similarly, the key plasmids to create the improved gRNA-marker insertions should be deposited to Addgene (or similar repository) and their ID numbers included in the resources table.

The plasmids have been deposited to Addgene and are currently being validated.

(14) The authors might consider including (perhaps as supplementary to Figure 1 or Figure 2) a crossing scheme for one of their MAGIC experiments. This will make it even clearer how a MAGIC experiment could be set up using existing fly reagents.

This is a good suggestion! We have added an example crossing scheme in Figure 1 – figure supplement 1C.